# Open Data Based Machine Learning Applications in Smart Cities: A Systematic Literature Review

**Luminita Hurbean** [1,*]**, Doina Danaiata** [1]**, Florin Militaru** [1]**, Andrei-Mihail Dodea** [1] **and Ana-Maria Negovan** [2]

[1] Business Information Systems Department, West University of Timisoara, 300115 Timisoara, Romania; doina.danaiata@e-uvt.ro (D.D.); florin.militaru@e-uvt.ro (F.M.); andrei.dodea98@e-uvt.ro (A.-M.D.)

[2] SC Flextronics Romania SRL, 300115 Timisoara, Romania; ana.duma@e-uvt.ro

\* Correspondence: luminita.hurbean@e-uvt.ro

**Abstract:** Machine learning (ML) has already gained the attention of the researchers involved in smart city (SC) initiatives, along with other advanced technologies such as IoT, big data, cloud computing, or analytics. In this context, researchers also realized that data can help in making the SC happen but also, the open data movement has encouraged more research works using machine learning. Based on this line of reasoning, the aim of this paper is to conduct a systematic literature review to investigate open data-based machine learning applications in the six different areas of smart cities. The results of this research reveal that: (a) machine learning applications using open data came out in all the SC areas and specific ML techniques are discovered for each area, with deep learning and supervised learning being the first choices. (b) Open data platforms represent the most frequently used source of data. (c) The challenges associated with open data utilization vary from quality of data, to frequency of data collection, to consistency of data, and data format. Overall, the data synopsis as well as the in-depth analysis may be a valuable support and inspiration for the future smart city projects.

**Keywords:** smart city; open data; machine learning; systematic literature review

## 1. Introduction

Urbanization is an important demographic mega-trend. With a pace of growth of urban population that is estimated to double until 2050, there is no doubt that "the future of the world's population is urban" [1]. In a rapidly urbanizing world, all the factors concerned/involved in bringing solutions for sustainable living in urban agglomerations shock hands, from academia, visionary companies to policy makers all over the world.

Based on the advanced digital technologies and their implementation in the cities especially in the last decade, smart city (SC) became one of the most important concepts of the new economy and one of the most researched topics. In the history of ideas, smart city is a relative newcomer [2], and the interest for this research field has grown and publications have intensified after 2008, as reported in a recent systematic literature review [3]. The quoted analysis also states the fuzziness of the SC concept, while there is not a generally agreed definition in the literature; the SC concept is frequently described as many-sided, multidimensional, complex, widespread, or fuzzy, while being used in inconsistent ways [2,4–6]. Having all these different opinions, we have decided to combine the 'intelligent city' perspective with the multidimensional view of the smart city in our research. The multidimensional aspect is related to the idea that a smart city should perform well in areas such as economy, environment, living, mobility, as well as people and governance.

In the last decade, among the technologies used to create smart applications for cities and their citizens, we can enumerate IoT, cloud computing, big data, analytics, and artificial intelligence. Acknowledging the fact that artificial intelligence (AI) for smart cities is a developing field of research and practice, a systematic literature review performed in

2020 indicated that AI is applied in many smart cities' areas such as education, security, transport, energy, environment, health, land use, and urban governance [7]. They conclude that learning-based AI, also known as machine learning (ML) has a greater potential to solve SC problems than rules-based AI. In this respect, we have chosen to investigate the 'intelligent city' fostered by the impulse generated by the development of machine learning applications.

As stated in [8], open data should be the support of the new economy because the core of a smart city consists of the creation and use of data to generate new services and to support decision making. In addition, according to [9], data is one of the three major pillars of the SC, along with technology and people. Taking into consideration the ITC opportunities and the emerged open data movement, one of the most relevant emerged SC related domains combines open data (OD) with machine learning (ML) applications.

Literature provides plenty of papers that review and analyze the employment of ML methods in the smart cities. Numerous papers on the ML applications for SC were found as well as papers arguing about the OD relevance for the smart cities. Considering the OD in relation to ML applications for smart cities, there have not found literature reviews that connect the three concepts of SC, ML, and OD. Thus, this paper aims to fill this gap and provides a state-of-the-art review of open data-based machine learning applications for smart city. To do so, we applied the systematic literature review methodology.

At first, we have scrutinized the existent systematic literature review on smart cities, machine learning, and open data. We went through twelve selected papers that combine the three main pillars that are the background of our research (see all referred papers in Table 1) and discovered that only one had a similar approach as a starting point [10]. While the conceptual framework is comparable, the mentioned research [10] did not target the ML applications but optimizing sustainability for SC. Other SLRs were analyzed at this point to decide on the added value that our paper could generate in the literature.

**Table 1.** Related SLR papers.

| Paper | SLR Scope/Focus |
|:---:|:---|
| [4] | "A systematic literature review on maturity models that assess the level of maturity for smart city projects" |
| [11] | "A systematic literature survey on software architectures for big data systems", with a few connections with SC |
| [12] | "IoT challenges in smart cities and provide the gap between the existing state-of-the-art IoT application on S" |
| [13] | "A comprehensive analysis of the literature on interoperability of SC data platforms" |
| [10] | "Analyze the link between the concepts of smart cities, machine learning techniques and their applicability" |
| [14] | "Systematically investigated the evolution of OGD research" |
| [15] | "the relationship between big and open data and how they relate to the broad concept of open government" |
| [16] | "Covers the revision of the studies related to air pollution prediction using machine learning algorithms based on sensor data in the context of smart cities" and concludes that "open data movement has increased the number of research works in the field of machine learning, especially in the prediction of air quality". |
| [17] | "Comprehensive survey that explores the application of graph neural networks for traffic forecasting problems", presenting also "a comprehensive list of open data and source resources for each problem" |
| [18] | "The challenges faced by smart cities and the key role data mining, machine learning and statistical methods can play to enable intelligent solutions for different applications" |
| [19] | "Systematically reviews the top 200 Google Scholar publications in the area of smart city with the aid of data-driven methods from the fields natural language processing and time series forecasting" |
| [7] | "Generates insights into how AI can contribute to the development of smarter cities". |

After analyzing the coverage area of the listed SLR type works, the main scope of our SLR was set and the research questions were formulated. The main objective is to investigate the particulars of open data-based machine learning applications for specific smart city areas. The following research questions were crystallized: (RQ1) Which learning

types and algorithms are used in open data-based ML applications for each of the six smart city areas (smart governance, smart economy, smart mobility, smart environment, smart people, smart living)? (RQ2) What are the sources used for open data? (RQ3) What are the challenges of open data utilization in ML applications for smart cities?

In this respect, the remaining of this paper is structured as follows. Section 2 includes a theoretical background covering the main concepts of SC, OD, and ML and Section 3 explains the research methodology. In Section 4, we present and discuss the results of reviewing the selected papers and data analysis using Power BI. The paper ends with Section 5 that includes research conclusions. In addition, Appendix A contains the list of the papers that were selected for the in-depth analysis.

## 2. Theoretical Background

### 2.1. Smart Cities

During the last decade, the concept of SC has evolved from the simple implementation of information technologies in the public services into an ecosystem that also takes into account innovation, the environment, the human, or the social aspects [4]. In [13], SC is considered not an ecosystem, but a development vision where ITC based solutions are integrated, data is acquired from heterogeneous sources, and assets are connected in a platform with the objective of improving life's quality while enhancing the efficiency and the economical value.

Based on a systematic effort of analyzing over one hundred SC definitions from various sources like: academic research community, government programs, different organizations (European Commission, United Nations, ITU etc.), corporations or standards development organizations, the focus group constituted within ITU proposed a comprehensive and integrative SC definition. In their regard, SC is "an innovative city that uses ICTs and other means to improve quality of life, efficiency of urban operations and services and competitiveness, while ensuring that it meets the needs of present and future generations with respect to economic, social, and environmental aspects" [20]. The same focus group has inventoried eight key aspects that support a sustainable smart city: "(1) quality of life and lifestyle, (2) infrastructure and services, (3) ICT, communications, intelligence and information, (4) people, citizen and society, (5) environment and sustainability, (6) governance, management and administration, (7) economy and finance, and (8) mobility" [20].

Closely related to our approach, [5] considers the SC being a city that employs technology to work toward the public problems "on the basis of a multi-stakeholder, municipally based partnership". Further, [5] has also formulated six characteristics or dimensions of the SC: (1) smart economy, (2) smart mobility, (3) smart environment, (4) smart people, (5) smart living, (6) smart governance. Actually, these characteristics actually represent the areas (domains) that SC initiatives focus on. They are described in Figure 1.

| Area | Description | Actions | |
|------|-------------|---------|---|
| **Smart Governance** | "... joined up within-city and across-city governance, (...) so the city can function efficiently and effectively as one organism." | • Promote online public services;<br>• Provide electronic voting;<br>• Promote e-government strategies; | • Provide transparent governance;<br>• Encourage citizen participation;<br>• Implement e-democracy. |
| **Smart Economy** | "... e-business and e-commerce, increased productivity, ICT-enabled and advanced manufacturing and delivery of services, ICT-enabled innovation ..." | • Deployment of ICT in business;<br>• Design strategies for economic growth;<br>• Attracting talent and promote creativity; | • Provide support for entrepreneurship;<br>• Develop business collaborations;<br>• Provision of tax payment systems. |
| **Smart Mobility** | "... ICT supported and integrated transport and logistics systems". | • Provision of international accessibility;<br>• Availability of safe transport systems;<br>• Traffic and parking systems; | • Availability of bicycles and footpaths;<br>• Provision of mobile internet;<br>• Provision of Wi-Fi hotspots in cities. |
| **Smart Environment** | "... smart energy including renewables, ICT enabled energy grids, metering, pollution control and monitoring, renovation of buildings and amenities, green buildings, green urban planning ...". | • Support pollution reduction;<br>• Provide environment protection;<br>• Recycling of solid waste;<br>• Provision of sewerage treatment; | • Provision of early warning systems;<br>• Provision of fire stations disaster alarms. |
| **Smart People** | "... e-skills, ICT-enabled working, access to education and training, within an inclusive society that improves creativity and fosters innovation". | • Presence of a university in the city;<br>• Existence of digital development in classrooms; | • Collaboration between knowledge centers;<br>• Plans for research, development and innovation. |
| **Smart Living** | "... ICT-enabled life styles, behavior and consumption". | • Promote electronic health policies;<br>• Provision of emergency response facilities;<br>• Provisioning 24/7 electric supply; | • Provision of 24/7 water supply;<br>• Guarantees safety and better housing;<br>• Provision of metering and online payment. |

**Figure 1.** The Smart City Areas (source: adapted from [5,21]).

The SC concept has gained greater attention since 2008 also due to the launch of the visionary IBM Smarter Planet project, where SC is defined as "a comprehensive approach to helping cities run more efficiently, save money and resources, and improve the quality of life for citizens" [22]. The expanding role of the data in the SC came slowly in the center of interest; in a report of the Academy of Smarter Communities, it is stated "IoT and data platforms play a central role ( . . . ) in managing the vast amount of data generated across different urban domains" [23]. More recently, Oracle, another visionary company, put forward their solutions to support cities to tackle high volumes of data—"combining artificial intelligence and machine learning to transform existing platforms into automated and mobile-friendly citizen services" [24].

### 2.2. Open Data for Smart Cities

In the recent years, the proliferation of technologies such as IoT and analytics, along with the constant growth of the data volume (big data) have motivated the vision of data and technology being used to create a better and sustainable quality of life of the citizens and businesses that inhabit the city. Based on the recent developments, we assert that cloud-based data and technology are used to make possible data-informed decisions in real time that improve the urban management. Overall, the recent literature agrees that data became a key feature in the smart city conceptualization and the new framework is now designed around three pillars: people, data, and technology.

Taking into account that a smart city initiative has to be developed around the data, [9] specifies that nowadays, it depends on connections, open data, and sensors. The nature of collected data depends on various factors and can vary from health services to governmen-

tal measures, social, economic, and environmental impacts [25]. For a long time, public organizations gather, manage, and process data for their internal operations. In the last decade, the emerged open data movement encouraged them to make their data available to the public as 'open data' [26]. Today, the ecosystem of a smart city has a plethora of sensors that generate large amount of data [27]. Aside of these sensors, data is also collected using different tools and technologies available, as follows: cameras, kiosks, personal devices, appliances, social networks and others. Data collection is a helpful tool, for both citizens and planner, helping to regain control and to access necessary information [28]. Data that are relevant for the SC can be gathered from numerous heterogeneous sources, from sensor data to user-contributed data in participatory sensing [29]. When we mention open data, we must understand that we do not limit this area just to government data, because the private sector also recognizes the potential benefits of sharing data under the umbrella of open data [30].

Today, both public and private entities appreciate the value of data because this resource has already proved to be the key to improving efficiency and effectiveness in everyday activities [31]. For the SC, the number of stakeholders is higher and more diverse than in the private business case, i.e., utility companies, transport providers, mobile phone operators, social media sites, financial institutions, surveillance and security providers, emergency services, and others, along with the citizens themselves [28].

In [32], open data (OD) is defined as data freely available to everyone to use and republish as they wish, without any restrictions (copyright, patents or other control mechanisms). The European Portal for open data states the following three features for open data: free flow of data, transparency, and fair completion. In [30] authors report that open data should comply with the following 10 principles: "complete, primary, timely, accessible, machine-processable, non-discriminatory, non-proprietary, permanent, license-free, and preferably free of charge." In the same respect, [33] mentioned that open data should be complete, primary (should include original data and metadata about data collection), timely and permanent (having appropriate control mechanism for data versions) and [34] stated the fact that openness is a good governance principle. However, open data proliferation has brought potential perils and insecurity, due to aspects such as who benefits from them and who might be harmed by data sharing [8].

The following categories of open data are acknowledged in [29]:

- Sensor data: data collected from different type of sensors found in a city (from traditional sensors that provide data about physical phenomenon to wearables that collect data about human activities and behavior);
- Image and video data: mostly data from video surveillance or other video sources;
- Text data: a complementary data source for many smart city applications.

As stated in [33], the most valuable sources of data are represented by open data initiatives of the government. In the last couple of years, these initiatives have burst out around the world, founding a goldmine for public administration [35]. Similar to open data, open government data (OGD) are available and accessible to everyone for their own needs, and they are made freely available for re-use for any purpose. The differentiation comes from the fact that they are produced with public money and the license specifies the terms of use (data.Europa.eu). For the reason of our research, we will further use the term open data, whilst it also refers to open government data.

Open data proliferation has established a 'data commons'. Similar to a park or a playground, the data commons is a public good, which is accessible to the public. Open digital data can be copied limitlessly while the original in physical terms is not affected in any way [8]. Open data or OGD, as a source of information and knowledge in a knowledge-based economy, might well be a free resource for end-users; however, its production, maintenance and gathering need to be secured and maintained, with significant cost by skilled staff, with appropriate AI and Big Data technologies, and through implemented systems with open standards [34].

Data is collected using different tools and mechanisms; therefore, we can find different type of data sources. Open data sources include any information that can be obtained without a privileged position [36]. The most relevant open data sources are: social media (Facebook, Instagram, Twitter, LinkedIn, YouTube, and other), electronic media (newspapers, news sites, other), blogs (Wordpress, Tumblr), booking and accommodation (Booking, FourSquare), satellite imagery (Landviewer, Copernicus Open Hub, Sentinel Hub etc.), and government data (World Bank Open Data, European Union Data Portal, open data in Canada, Data.gov, country level sites with open access to government data, and many more).

At the governmental level, open data can be a powerful force for public accountability, as [37] mentions, because information can be analyzed, processed, and combined in an easier way, which allows a new level of public scrutiny. Making data available for public will increase governmental engagement from citizens and potentially add value to the data [38]. Data availability definitely supports innovation and contributes to economic growth.

There are data sets available, which allow direct access to data and so, interested parties gain instant and easy access to data. However, there are situations in which there is the need to perform data extraction from available data sources using different techniques. What is important for any application is the quality of gathered data, in order to have a correct representation of the real world and to be fitted for their intended use [31]. Furthermore, the following aspects are also extremely relevant for developing applications for smart cities [39]: (1) storing and managing databases, as large amount of data is collected, and (2) integrating data from many sources.

*2.3. Artificial Intelligence for Smart Cities: Machine Learning and Deep Learning*

Moving forward in the SC ecosystem, another main component is technology; after 2010, more and more different technologies have been employed in smart city related developments. Information and communications technologies enable the detection and collection of data, diffusion of the data through the network and development of specific applications. In the recent years, key domains, such as urban planning, transportation, or energy make use of new technologies to provide smart applications to cities and their people: networking and communications, IoT, big data, analytics, cloud or edge computing, or artificial intelligence [40]. Other new technologies adopted in the SC context are autonomous vehicles, 5G, blockchain, virtual reality, and digital twins [41].

IoT (short for Internet of things) is a tool that provides specific services that give low level support to different applications offered to citizens. Technological advances, such as standard communication protocols and wireless networks made it possible to obtain sensor data at any time and everywhere. The cloud-based infrastructure of a smart city architecture allows the information to be communicated to the connected objects/entities. The cloud offers the adjustment of computational resources according to the demand and transforms capital expenditures into operational costs. The enormous production of digital data in cities is due to the fact that all the actions effected on the personal computers, laptops, mobile phones, and other connected objects leave a trace. Big data offers the capabilities to capture, sore, manage and analyze this huge volume of information, which is persistently growing, accumulating, and waiting to be analyzed [42].

Artificial intelligence (AI) represents an innovative technology meant to deal with the urban challenges of environment, people, transportation, security, or economy. Even more, AI is a key enabler to improve data processing and transformation into useful information and knowledge intelligence for the sustainable cities [21]. While initially being defined as the science of making machines intelligent, today's AI represents a combination of machine learning and deep learning techniques. Things have evolved a lot in less than ten years from the moment when the Google's unsupervised neural network learned to recognize cats in YouTube videos with 74.8% accuracy. In the SC environment, where types of data acquired vary from text, images, videos, social media, or sensors, AI has the potential

to analyze the gathered and integrated big data and employ cloud computing for the operational costs and resources optimization [43].

Machine learning (ML) allows applications to become more precise at predicting outcomes without being explicitly programmed to do so. Their algorithms use existent data as input and learn to predict new output values. Based on how algorithms learn, there are four ML types: supervised learning, unsupervised learning, semi-supervised learning, and reinforcement learning (see Figure 2 for an overview of each type along with the problems they can solve).

| ML type | How it works | Problems they can solve |
|---|---|---|
| **Supervised learning** | The algorithm is trained with labeled inputs and desired outputs. | • Classification (binary/multiple): choose between two or more types of answers. <br> • Regression based (Bayesian) modeling: predict of continuous values. <br> • Ensembling: combine predictions of multiple ML models to get an accurate prediction. |
| **Unsupervised learning** | The algorithm examines unlabeled data to look for patterns, which are used to create data subsets. | • Clustering: split a data set into groups based on similarity. <br> • Anomaly detection: identify unusual data points in a data set. <br> • Association mining: identify sets of items in a dataset based on occurrence. <br> • Dimensionality reduction: reduce the number of variables in a data set. |
| **Semi-supervised learning** | The algorithm is trained with a small number of labeled data and then works with new, unlabeled data. | • Machine translation: teach algorithm to translate based on less than a full dictionary. <br> • Fraud detection: identify cases of fraud based on a few positive examples. <br> • Labelling data: algorithm trained on small data set learn to apply data labels to larger sets. |
| **Reinforcement learning** | The algorithm is programmed with a specific goal and a given set of rules for accomplishing that goal. | • Robotics: robots can learn to perform tasks in the physical world. <br> • Video gameplay: teach bots to play video games. <br> • Resource management: with finite resources and a defined goal, it help enterprises to allocate resources. |

**Figure 2.** ML types: how they work and what problems can they solve (source: adapted from [44,45]).

In supervised learning, algorithms are supplied with labeled data to be used in training (input) and variables that the algorithm assesses for correlations are defined (output).

Unsupervised learning employs algorithms that train on unlabeled data. The algorithm is capable to find meaningful connection(s) in the scanned data set. The trained data and the predictions are predetermined.

Semi-supervised learning involves a mix of supervised and unsupervised. While the algorithm is fed with mostly labeled training data, the model has the liberty to investigate other data. The result is based on its own understanding of the data set.

Reinforcement learning teaches a machine to perform a "multi-step process" based on plainly defined rules. While the algorithm is programmed using positive or negative indications to fulfill a task, it may also determine on its own what steps to take during the process.

The last decade has been the decade of neural networks (also known as ANN—artificial NN) due to the availability of the computational power and the data required for good training. Algorithms and architectures were adapted to the neural networks specifics. Imitating the human brain behavior, a NN includes node layers with definite roles: one input layer, one or more hidden layers, and one output layer. Nodes are connected to one another and they also have an associated weight and threshold. From a node, data is sent to the next layer only if the output of that node is above the indicated threshold value. Otherwise, no data is passed along to the next layer [46]. The NN that includes more than three layers is considered a deep learning algorithm (see Figure 3).

Neural networks provide a multitude of advantages: they require less formal training because of their excellent learning capabilities, they are able to detect complex nonlinear relationships, they may work with manifold different training algorithms, and they prove their flexibility because they understand various forms of data [47]. On the other hand, NN algorithms necessitate more time and large computational operations to train a model with large volumes of data and their ability to explicitly identify causal relationships between variables is limited. A great challenge for NN is to avoid overfitting, which impedes the NN capacity to generalize well to new data. In this respect, NN algorithms perform well when their complexity is fitting the complexity of the data.

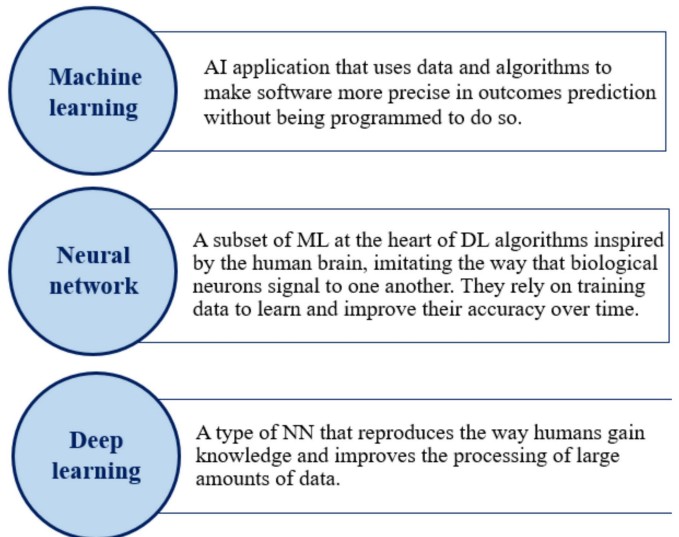

**Figure 3.** ML versus NN versus DL (source: adapted from [46,48]).

Deep learning (DL) is considered a subset of ML but is different in the way algorithms learn and how much data each type of algorithm uses. It is very beneficial that DL eliminates manual intervention through the automation of much of the feature extraction part of the process. Being able to use large data sets, it earned the title of "scalable machine learning". From a practical point of view, deep learning is meant for more complex use cases because a DL model requires more data points to improve its accuracy, while machine learning is able to work with smaller datasets because a ML model may rely on less data.

As also reported in [10], in the last five years there has been an exponential growth for articles with experiments or applications of machine learning techniques in all the smart city areas. Different from them, we are taking into consideration the immense interest for the open data platforms that are constantly expanding in the public space and, in view of that, we will investigate the specifics of open data based machine learning applications for smart cities.

## 3. Methods

This section describes the methodology applied for the systematic literature review. The process of systematic literature review consists of the following activities: formulate research questions, select studies, extract required data, analyze and synthesize data, describe the results. At first, we defined the research questions—clear statements that conduct our literature review. For the SLR methodology, we have used PRISMA, which consists of four phases: identification, screening, eligibility, and included, together with a comprehensive checklist. The included results were then assessed and interpreted to give answers to the research questions.

### Research Questions

RQ1. Which learning types and algorithms are used in open data-based ML applications for each of the smart city areas?

RQ2. What are the sources for open data?

RQ3. What are the challenges of open data utilization in ML applications for smart cities?

**Search Strategy and Criteria for Inclusion/Exclusion**

In our search strategy, we have started with selecting major scientific databases, i.e., Web of Science, Scopus, IEEE eXplore, AIS (Association for Information Systems library), Springer, and ProQuest, along with some popular ones such as Semantic Scholar and MDPI. We have tracked published research results in June 2021 using a comprehensive search string:

*(("artificial intelligence" OR "machine learning" OR "deep learning") AND ("Open Data" OR "open government data") AND ("smart city" OR "smart cities"))*

A total number of 472 papers, all written in English, from 2011 to 2021 period were selected in the eight searched databases (see Table 2).

**Table 2.** Papers found/database journal.

| Database Journal | Papers Found |
|---|---|
| Web of Science | 53 |
| Scopus | 45 |
| IEEE eXplore | 63 |
| AIS | 19 |
| Springer | 113 |
| Proquest | 13 |
| MDPI | 58 |
| Semantic Scholar | 108 |
| TOTAL | 472 |

According to PRISMA method, the initial 472 records were screened with the first purpose of eliminating duplicates. Consequently, 61 records were excluded and the explanation resides in fact that papers are frequently indexed in more than one database. The 411 obtained records were further screened and inclusion and exclusion criteria were applied in two rounds (see Figure 4). In the first round, papers were checked for eligibility to make sure they were peer-reviewed articles and that they discuss machine learning applications for smart cities and open data. Assessing their abstracts, 194 papers were excluded. The large number of excluded papers is the consequence of having a long list of papers based on the comprehensive search string. When carefully reading the abstracts, we found out that many papers did not actually make use of open data or did not employ machine learning techniques for smart city applications and therefore were eliminated.

In the second round, the remaining 187 papers were examined in their content (in-depth analysis) in order to retain for evaluation only the papers with actual applications (experiments), which were elaborated with machine learning techniques, and based on open data. We excluded 118 papers because they did not have definite applications that were developed with ML techniques—they were position papers, present frameworks or taxonomies, or some models. Only 69 eligible papers were included in the systematic review that was performed through in-depth analysis of: used ML techniques, type of open data and the challenges encountered in data utilization, and the SC area that the application addresses.

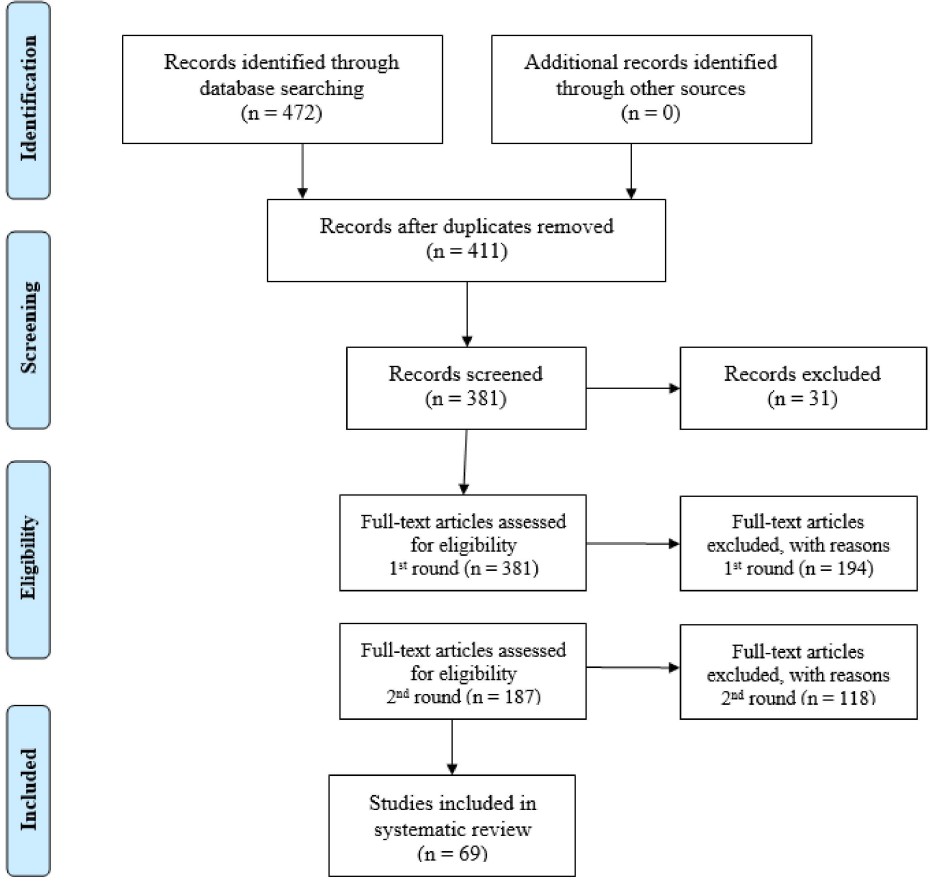

**Figure 4.** The PRISMA diagram.

## 4. Results and Discussion

Our relevant sample includes 69 selected records, out of which 43 are published journal articles and 26 in the proceedings of international conferences. Appendix A includes the complete list of selected papers with a synthetic description for each of them. While the search was run for period 2011–2021, after the screening of the records, it has reduced to 2013–2021 due to the lack of papers published in 2011–2012. Data was initially collected using a shared Google sheet and then it was exported and processed to obtain visualizations with Microsoft Power BI. Results are detailed as follows.

As regards the time analysis, the records cover a period of 9 years, starting from 2013 and the latest ones were published in 2021. Figure 5 pictures the distribution in time. The trend indicates a significant growth starting with 2017 with a maximum in 2019. It is our belief that the ascending trend illustrated for 2017–2020 will continue given that in the last couple of years open data initiatives favored data-driven innovation [49] and fostered the delivery of ML based smart solutions. The fact that we have only five records in 2021 is attributable to the searching time (June 2021); as a consequence, we cannot have a final number for the papers published in 2021.

When investigating the areas of SC application as we have previously described (see Figure 1), only two areas are slightly represented, i.e., smart people (two applications) and smart economy (four applications); the total and the specific articles (coded as in Appendix A) are presented in Figure 6.

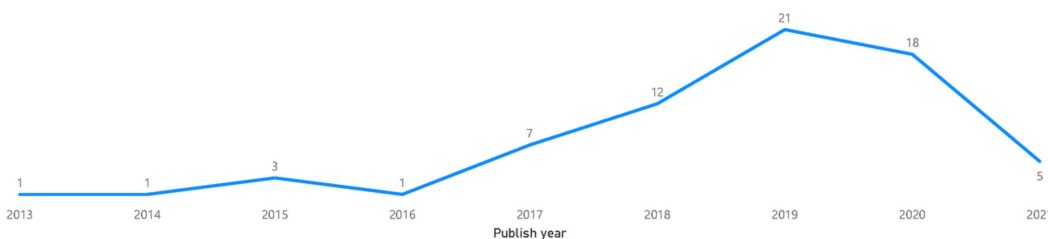

**Figure 5.** The records' distribution in time.

| | | |
|---|---|---|
| Smart Economy | [S12]; [S31]; [S46]; [S49]; | 4 |
| Smart environment | [S7]; [S16]; [S19]; [S25]; [S27]; [S30]; [S36]; [S38]; [S39]; [S45]; [S54]; [S57]; [S60]; [S64]; [S65]; | 15 |
| Smart governance | [S1]; [S2]; [S3]; [S11]; [S14]; [S17]; [S24]; [S35]; [S37]; [S40]; [S41]; [S47]; [S59]; [S62]; [S67]; [S68]; | 16 |
| Smart living | [S4]; [S5]; [S10]; [S13]; [S23]; [S29]; [S32]; [S34]; [S42]; [S48]; [S50]; [S61]; [S69]; | 13 |
| Smart mobility | [S8]; [S9]; [S15]; [S18]; [S20]; [S21]; [S26]; [S28]; [S33]; [S43]; [S44]; [S51]; [S52]; [S53]; [S55]; [S56]; [S58]; [S63]; [S66]; | 19 |
| Smart people | [S6]; [S22]; | 2 |
| **Total** | | **69** |

**Figure 6.** List of papers classified by smart city areas considered in proposed applications.

*RQ1. Which Learning Types and Algorithms Are Used in Open Data Based ML Applications for each of the Smart City Areas?*

Machine learning algorithms offer a world of potential, making available to developers many routes to take, along with the type of machine learning they opt for, in a wide variety of smart city applications.

As revealed above, use of machine learning with open data in SC is a recent topic of interest; we only found papers dating from 2013 to 2021. While at first supervised learning stood out as the preferred ML technique, in the last five years deep learning has been the definite most often used machine learning type in open data-based SC applications (see Figure 7). This may offer valuable insight in terms of the approach to take when designing open data-based SC solutions. Given the nature of imperfect data within open data sources and the seemingly random data points generated within a city, deep learning may be the most relevant tactic to use for such circumstances. Besides, unlike traditional ML algorithms, DL can deal with great amounts of data, therefore, providing high-level solutions to the smart city problems [49].

Based on this reasoning, we wanted to visually signify the distinction between classical machine learning and neural networks/deep learning. The classical ML algorithms were divided into the following four categories: supervised learning, where data is labeled (e.g., decision trees, or linear regression); unsupervised learning, where data is not labeled (e.g., K-means, SVD), semi-supervised learning, where some of the data is labeled and some is not, and reinforcement learning. Deep learning is the fifth category, and the grouping of deep learning algorithms (such as LSTM or CNN) was focused on our assessment of the algorithms that best fit the NN rendition of hidden layers.

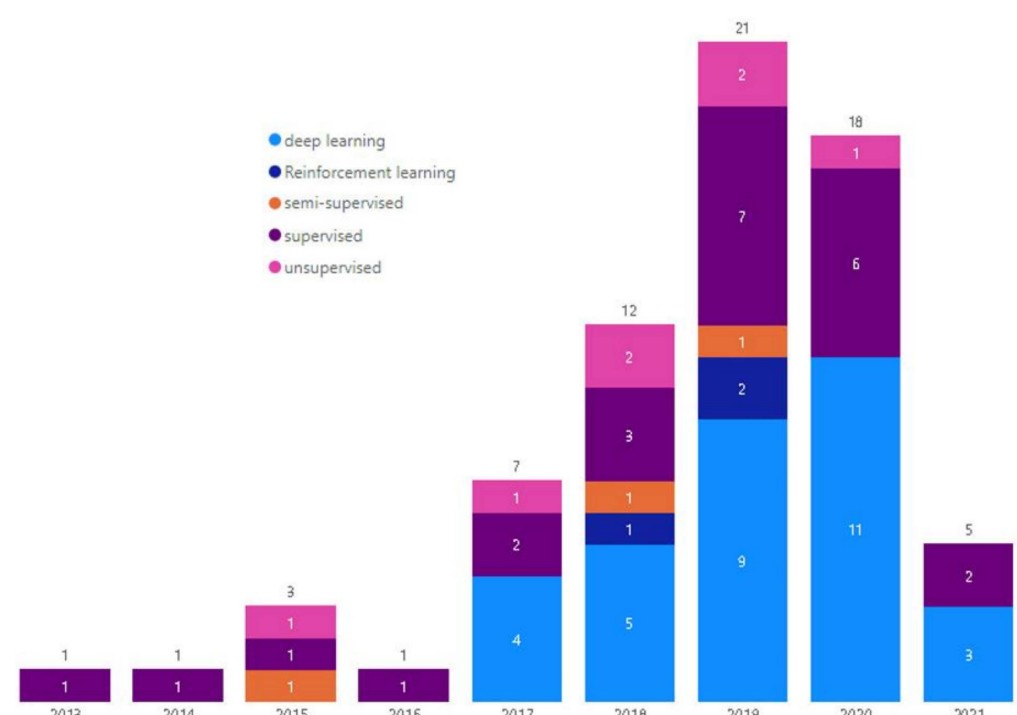

**Figure 7.** Machine learning techniques used in applications classified by papers' publish year.

The complete list of papers organized for each of the five ML types discovered in our sample (coded as in Appendix A) is presented in Figure 8.

| deep learning | [S1]; [S5]; [S6]; [S7]; [S12]; [S13]; [S14]; [S15]; [S16]; [S17]; [S23]; [S24]; [S29]; [S30]; [S31]; [S33]; [S34]; [S36]; [S38]; [S39]; [S41]; [S42]; [S44]; [S45]; [S46]; [S54]; [S58]; [S61]; [S63]; [S64]; [S65]; [S66]; | 32 |
|---|---|---|
| supervised | [S3]; [S4]; [S8]; [S10]; [S11]; [S18]; [S19]; [S20]; [S21]; [S25]; [S28]; [S35]; [S37]; [S47]; [S48]; [S49]; [S50]; [S51]; [S55]; [S57]; [S60]; [S62]; [S67]; [S68]; | 24 |
| unsupervised | [S2]; [S22]; [S27]; [S40]; [S43]; [S56]; [S69]; | 7 |
| Reinforcement learning | [S26]; [S52]; [S53]; | 3 |
| semi-supervised | [S9]; [S32]; [S59]; | 3 |
| **Total** | | **69** |

**Figure 8.** List of papers classified by the machine learning techniques utilized in applications.

Overall, three of the machine learning techniques stand out: deep learning (46.38% of the papers), supervised learning (34.78%), and unsupervised learning (10.14%). The reasoning here is that supervised learning is constantly used in SC applications because it has proved its value, while deep learning arose recently but has proved to be more suitable for the SC applications. Considering on deep learning-based applications, we dove deeper in order to identify what algorithms are mostly used (see Figure 9) and the results include long short-term memory (LSTM), convolutional neural networks (CNN), and artificial neural networks (ANN), along with comparisons between different algorithms (multiple algorithms). The rest of the algorithms are used few times (less than two times) or are iterations of the aforementioned algorithms. From this finding, we may learn that so far in SC applications dominate LSTM, CNN and ANN algorithms.

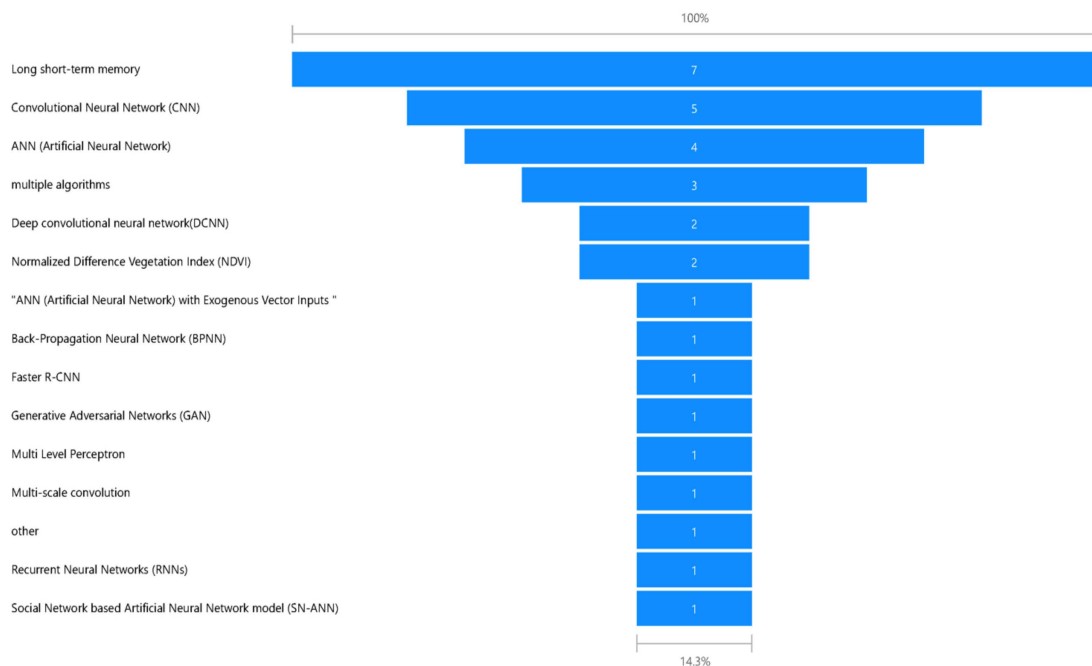

**Figure 9.** The ML techniques and algorithms used for deep learning applications.

As regards the types of learning employed in the deep learning applications, when classifying the algorithms they used (based on [50]), we have learned that 26 papers (37.68% of the total papers and 81.25% in the deep learning papers) applied supervised learning. Only one paper applies unsupervised learning and the other five are using hybrid or semi-supervised learning.

Another interesting piece of evidence is that in many ML applications researchers choose to use multiple algorithms; this is the case mainly for the supervised learning applications (54% of the cases), where the research focuses on identifying of the best choice of algorithm from a range of algorithms already confirmed for the specific application. The distribution of algorithms used in supervised learning SC applications is represented in Figure 10.

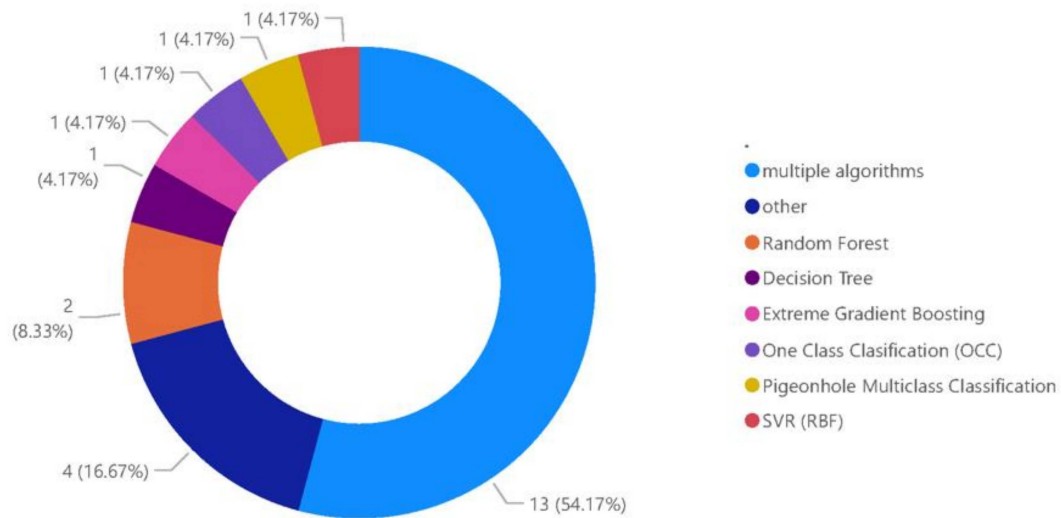

**Figure 10.** Distribution of algorithms in the papers using supervised learning.

Regarding the purpose of the applications, we investigated the SC areas relative to the ML techniques employed (see Figure 11 for all the data) and we have discovered the following:

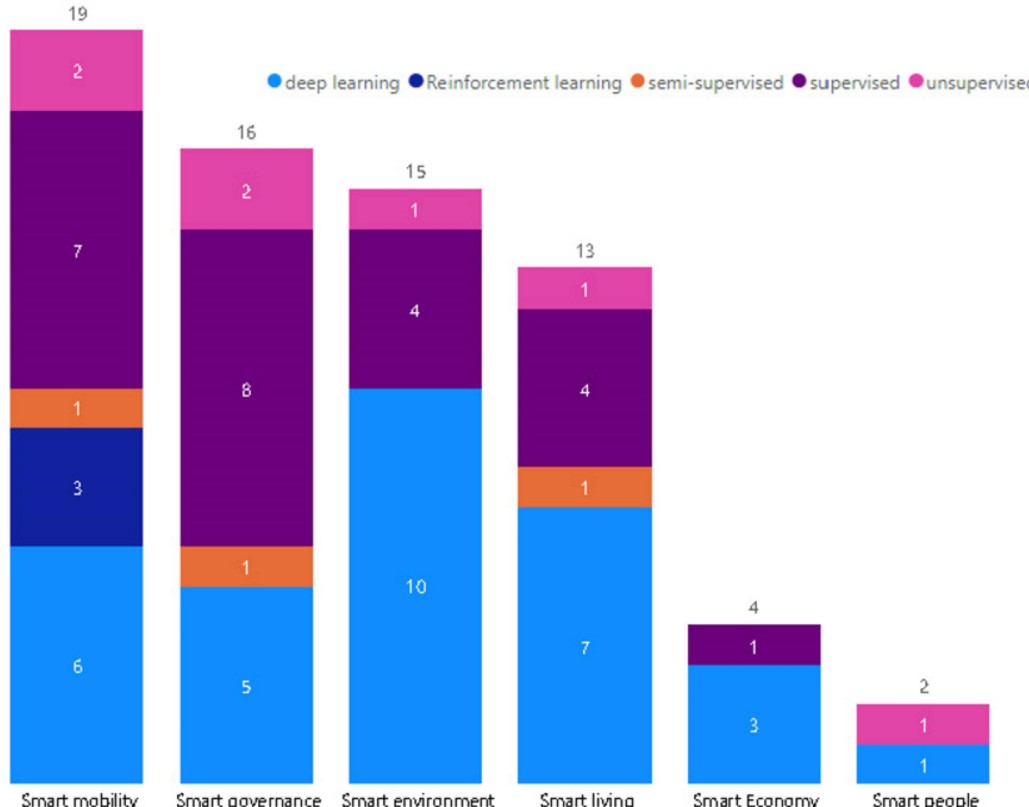

**Figure 11.** Machine learning techniques used in each of the smart city areas.

- Deep learning and supervised learning are reported in all the SC types of applications (excepting Smart people for one of them).
- Reinforcement learning has only one field of application, i.e., smart mobility. This looks similar to the 'multi-step process' but does not fit the other SC areas of application.
- There is an evident supremacy of deep learning application in half of the SC areas: smart economy (75% of the applications), smart environment (67% of the applications), and smart living (53.85%).
- Supervised learning is utilized in half of the smart governance applications.
- Even if it does not represent a first choice, we believe that unsupervised learning has a great potential for SC applications, and we have discovered one or two applications in almost all SC areas.
- If deep learning is the most prevalent machine learning type, semi-supervised learning is the last choice in SC applications.

*RQ2. What Are the Sources for Open Data?*

Smart city applications are growingly present in cities around the world, and these are designed to address different issues/topics for a modern city that is in continuous development. The trend of open data and open government can be validated again with the use of open data for developing smart city applications. This has been possible because over the last three decades, data sources have become available to the public once with the "open movement". The trigger for this movement has been the launch of Internet in 1991. There are several issues related to this topic, as both public and private organizations have adhered to it, but it has not been an impediment to make data available for the wide public. Data is collected from different sources, such as sensors, cameras, mobile devices

and many others, and that this data is found in different repositories, such as data sets, public websites, data platforms, and many others.

Researchers together with smart city applications developers need to define their scope and to clearly identify what data is needed for their purpose. Obtaining data is not the 'show stopper' anymore, as they have numerous options of choice.

Nowadays, open data are available from varying areas of interest making it possible to create different SC applications. While reviewing our sample of records we have found that 55% of the applications have used open data platforms (see Figure 12). This option is obviously explained through the availability of data at any time. It is not unexpected, considering that for each year that passes more and more governments and local authorities align with the "open policy" (providing data for a transparent administration).

| | | |
|---|---|---|
| Open data platform | [S1]; [S2]; [S5]; [S7]; [S8]; [S11]; [S13]; [S14]; [S18]; [S20]; [S21]; [S22]; [S23]; [S24]; [S26]; [S27]; [S28]; [S30]; [S31]; [S33]; [S36]; [S37]; [S40]; [S44]; [S47]; [S48]; [S49]; [S55]; [S56]; [S59]; [S60]; [S61]; [S63]; [S65]; [S66]; [S67]; [S68]; [S69]; | 38 |
| Multiple | [S10]; [S12]; [S15]; [S19]; [S25]; [S32]; [S41]; [S46]; [S58]; | 9 |
| Sattelite images | [S3]; [S4]; [S16]; [S17]; [S34]; [S35]; [S50]; [S54]; [S62]; [S64]; | 10 |
| Other | [S6]; | 1 |
| public websites (private companies) | [S9]; [S29]; [S38]; [S39]; [S42]; [S43]; [S45]; [S51]; [S52]; [S53]; [S57]; | 11 |
| **Total** | | **69** |

**Figure 12.** List of papers classified by the sources for open data used in ML applications.

If 30 years ago this was unimaginable, today open data platforms are normal commons. In our sample of records, we observe year 2015 as the first appearance of the open data platform as a source—there were two articles proposing smart city applications based on OD. The same number of two articles was reported for 2016 and 2017 and the number grows significantly in the following years. For example, of all articles of 2019 more than half (57%) make use of open data platforms, indicating a higher interest from the researchers to exploit this source of data.

Public websites of private companies are another source of data identified in our review (16%). There is a lot more effort needed to extract accurate data in this scenario, but it proves to be a valuable source of data. The mentioned effort refers to extracting, cleaning, and preparing the data. Numerous websites hold useful information that is shared with the public and that can be used in a variety of ways to build application for smart cities.

It is imperative for cities to implement more and more solutions that can support the citizens in terms of services provided by local authorities (example: authorizing new constructions and demolitions), but also in terms of quality of life (example: forestation of cities, pollution of areas, traffic control, etc.). This can be performed using other sources of OD, such as satellite images. This data source is reported in 15% of the reviewed articles. Considering the development of the satellite technology, their images have become more accessible to obtain, and there is no wonder that smart cities applications have found use cases.

Nevertheless, there are also situations in which multiple data sources are needed to obtain the desired result, case of complex projects or applications. In our review, this was the case for 13% of articles. We have encountered combinations such as the OD platform and user-generated, OD platform and private companies' websites, satellite images and user-generated, private companies' websites and private companies' websites. When using different type of data there are also accompanying difficulties, especially when the need to integrate data retrieved from numerous data sources in one dataset.

As regards the type of the smart city applications, it is noteworthy to observe that for all of them open data platforms represent the most utilized OD source (see Figure 13). Not surprisingly, applications for smart governance rely mostly on OD platforms with 69% of them using this source. In addition, 63% of the applications for smart mobility are using open data platforms.

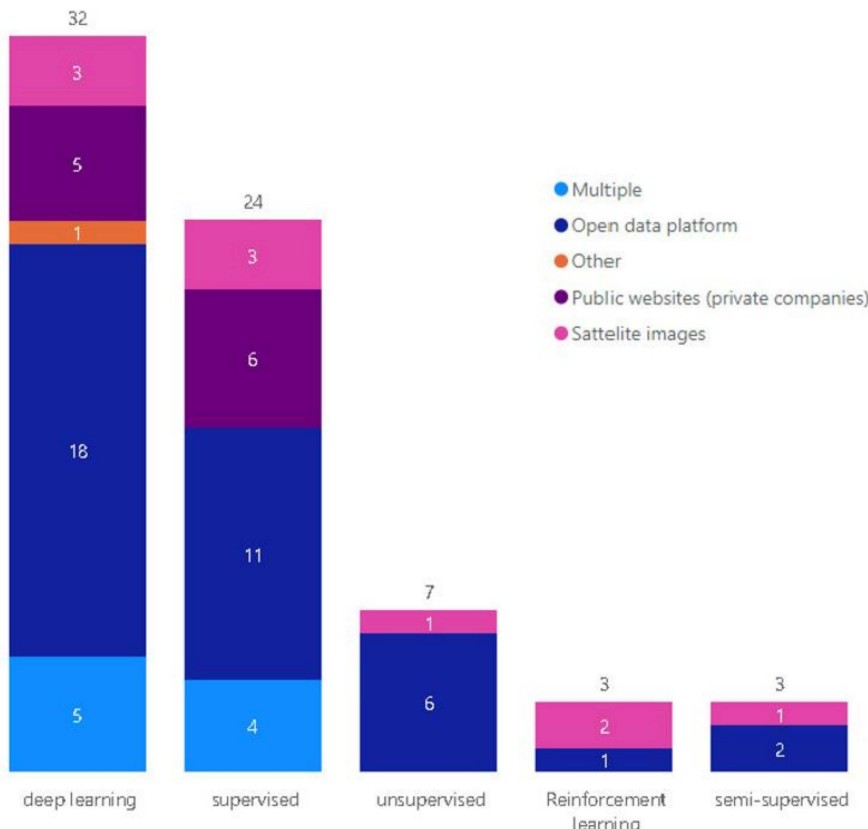

**Figure 13.** Open data sources used in applications for each of the ML techniques.

As regards the type of the smart city applications, it is noteworthy to observe that for all of them open data platforms represent the most utilized OD source (see Figure 14). Not surprisingly, applications for smart governance rely mostly on OD platforms with 69% of them using this source. In addition, 63% of the applications for smart mobility are using open data platforms.

We have determined interesting results when analyzing the types of data used for different ML techniques (see Figure 13). Regarding the OD platforms, they are the most used source of data in applications based on unsupervised learning, but they are also the favorite choice for supervised learning (58%) and deep learning applications (50%). The satellite images are used only in supervised learning and deep learning applications, while public websites data are useful for all the applications we have analyzed.

*RQ3. What Are the Challenges of Open Data Utilization in ML Applications for Smart Cities?*

In the modern society, the vast amount of data is a challenge in itself. On one hand, smart cities initiatives may take advantage of the large volume of available raw data and on the other hand, data collecting and sharing among all stakeholders is not an easy task. The open data movement—a new form of democracy—changed the circumstances and gave new prospects to the applications for smart cities. Shared open data portals ensure that all stakeholders are on the same page when information is updated effortlessly. When data portals open out from internal sharing to external publishing, inter-organizational 'synergy and connectivity' is attained in areas such as electricity, water, environment, traffic management, or safety.

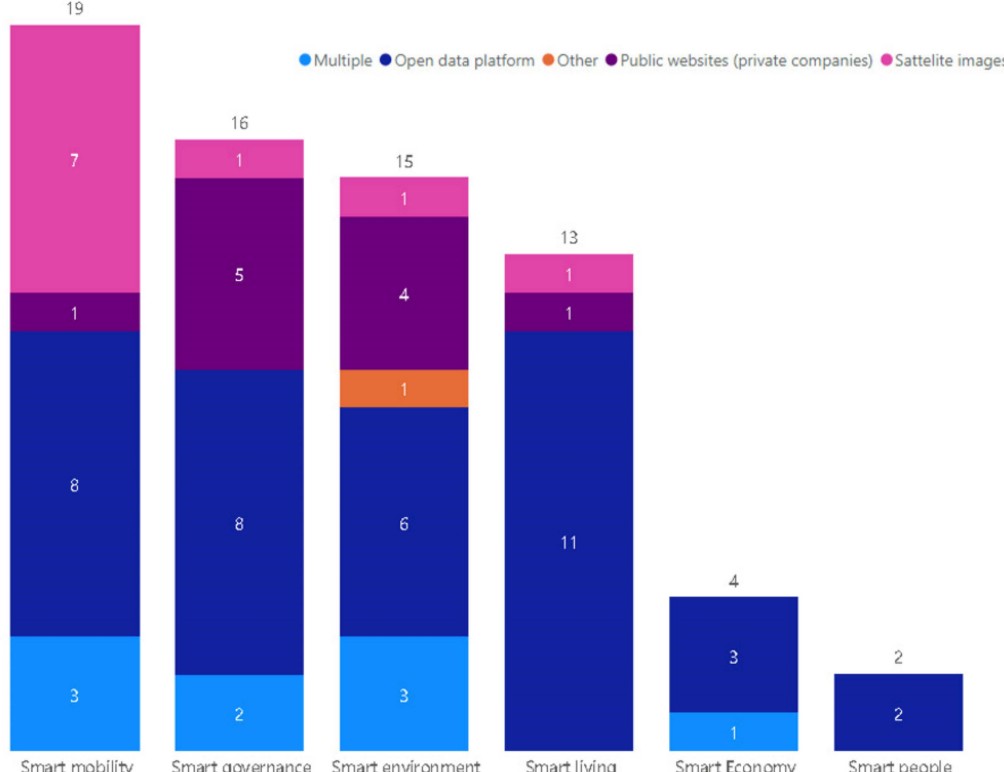

**Figure 14.** List of papers classified by the sources for open data used in ML applications.

The most frequent reported challenges for open data usage may be summarized as:

- Integrating data from many sources, as a result of having many heterogeneous sources of data (from sensors to social media) that originate from different public organizations/departments or even private companies;
- Multitude of data formats, with text data that is structured or semi-structured, but also images, videos and other unstructured data sources—they need to be harmonized in the data gathering phase;
- Quality of the data regarding aspects such as accuracy, consistency or data imbalance, sometimes affecting the data validation activity;
- Data traceability, using specific mechanisms to track the origin of the data based on accurate and reliable metadata.

In light of this, considering the third RQ, in the whole analyzed studies we have discovered that the most frequent entry (17 papers) was that no problem related to open data were reported. For those articles where problems were encountered, the challenge to overcome the multitude of data formats appears the most frequently (13 papers), followed by the challenge of not having all the needed data (11 papers), data quality issues (7 papers) and data consistency (5 papers). Another finding is that 13 of the developed applications come across with multiple problems related to open data utilization. The less confronted problems are having duplicates and having an imbalanced dataset (see Figure 15 where the results are presented in descendent order as percentages in the total number of papers).

With respect to the different sources of data utilized, some of the researchers reported that open data platforms are suitable as data source in their ML based applications (10 papers), while others confront with problems such as data format (nine cases) or insufficient data (nine cases) using the OD platforms source. In addition, most of the multiple problems situation was also encountered when using OD platforms (for 10 cases). When using the satellite images, the most reported challenges in data utilization are related to data quality and data format or consistency. The complete image on the confronted problems is represented in Figure 16.

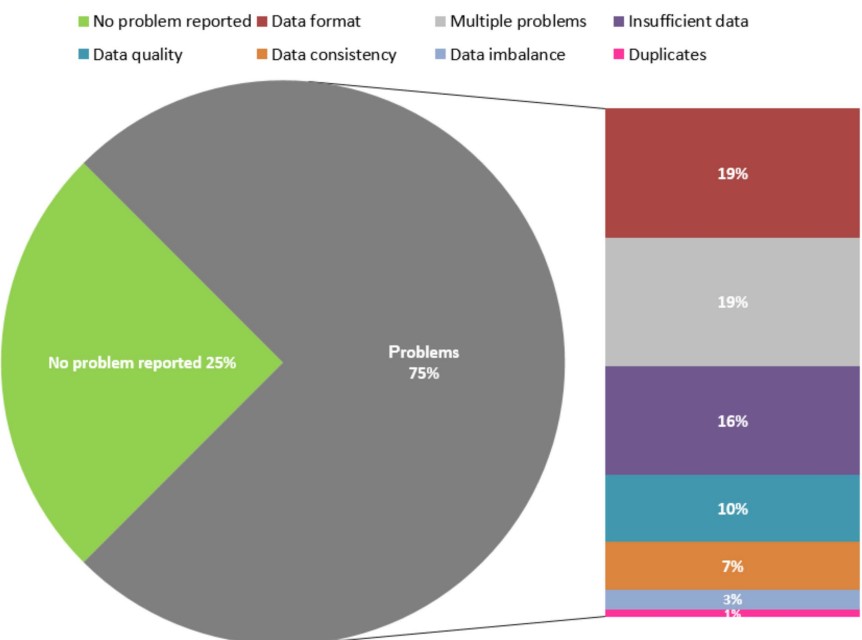

**Figure 15.** Frequency of challenges in open data utilization for all papers.

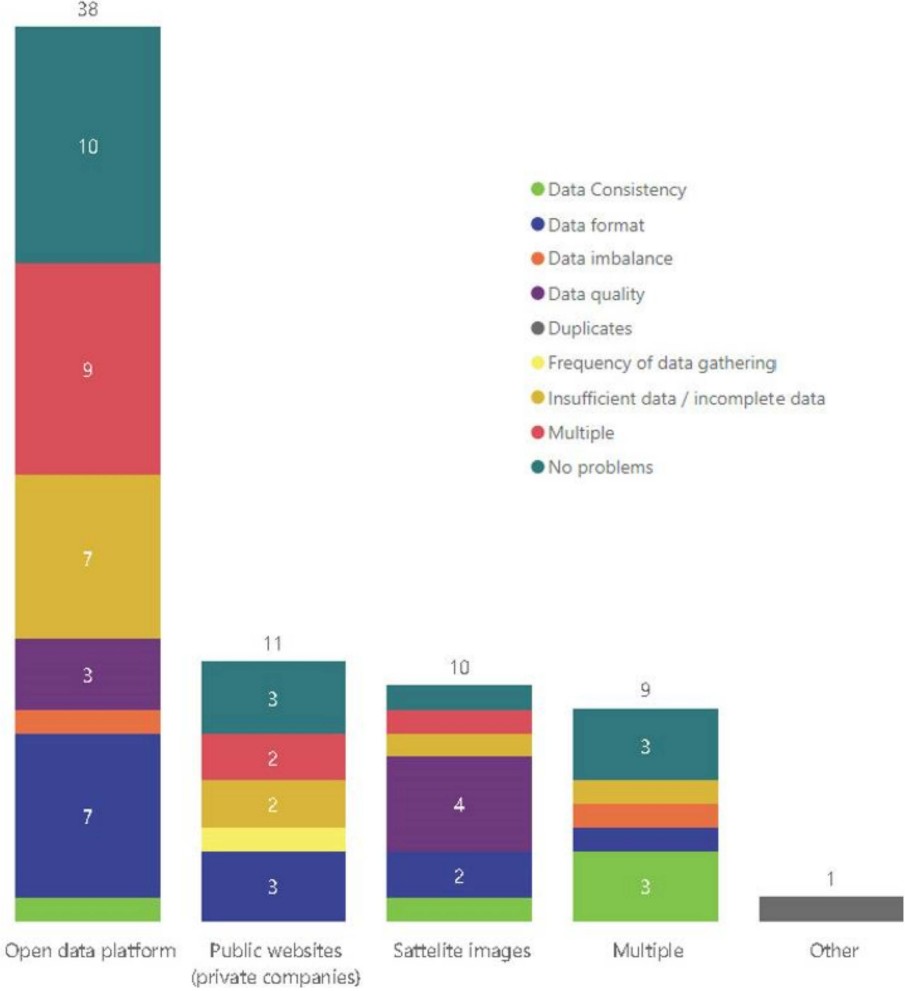

**Figure 16.** Challenges in open data utilization for each of the OD sources.

Mining into our dataset with the challenges meet for different types of machine learning applications, we have discovered 25 out of the total deep learning based applications (meaning 78%) have encountered problems related to data utilization. The most frequently, researchers come across the problem of not having sufficient data or data being incomplete (25% of the DL application papers) and have to address the problem with the data format (19% of the DL application papers). The situation looks better for ML applications using supervised learning where in almost a third of them there were no problems with data utilization. For this type of application, frequent problems were associated to data format (17%) and data quality (17%) and likewise, frequently, researchers indicated that they have met multiple problems with data utilization (17%). Figure 17 depicts the complete report on challenges related to data utilization for each type of ML application.

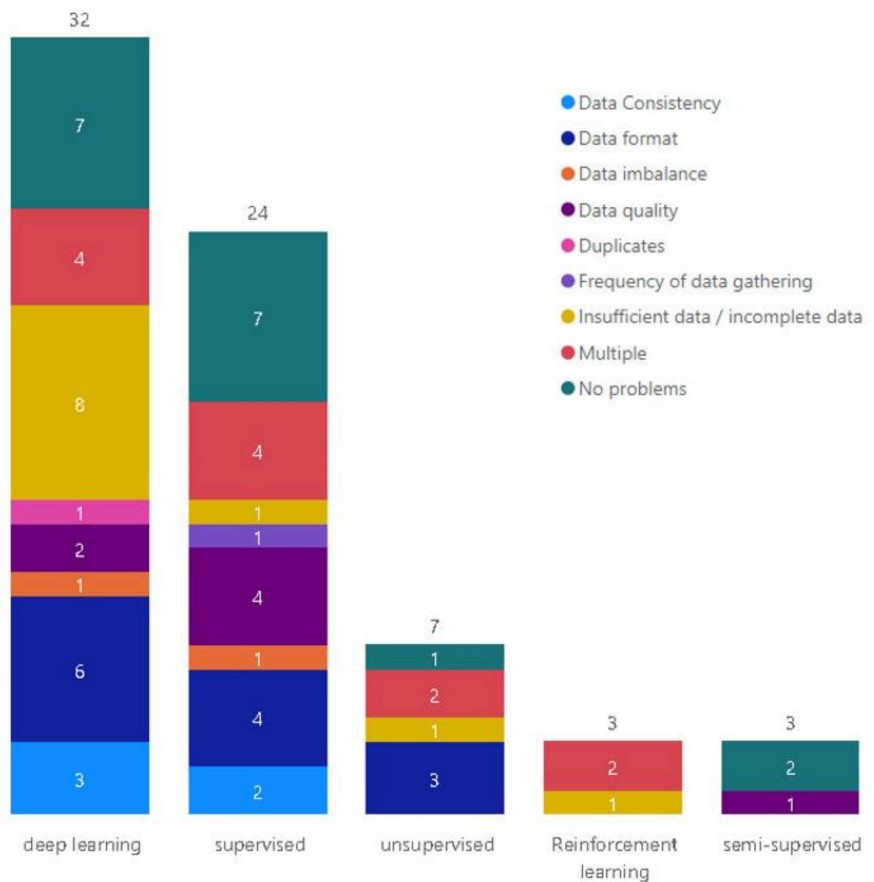

**Figure 17.** Challenges in open data utilization for each of the ML techniques.

With respect to the SC areas of the applications, we have observed that data format is a challenge for applications in smart living (38%) and smart mobility (25%), while applications in smart living also confront the problem of not having sufficient or complete data. All results of the analysis of open data utilization challenges for different areas of SC applications are included in Figure 18.

The area of SC where we met most of the applications (31%) that have not encountered problems with open data utilization is smart governance. Regarding this opinion of 'no problems', we should mention that possibly some authors have not included a discussion on the problems with open data utilization. Our results are based on the reported problems only.

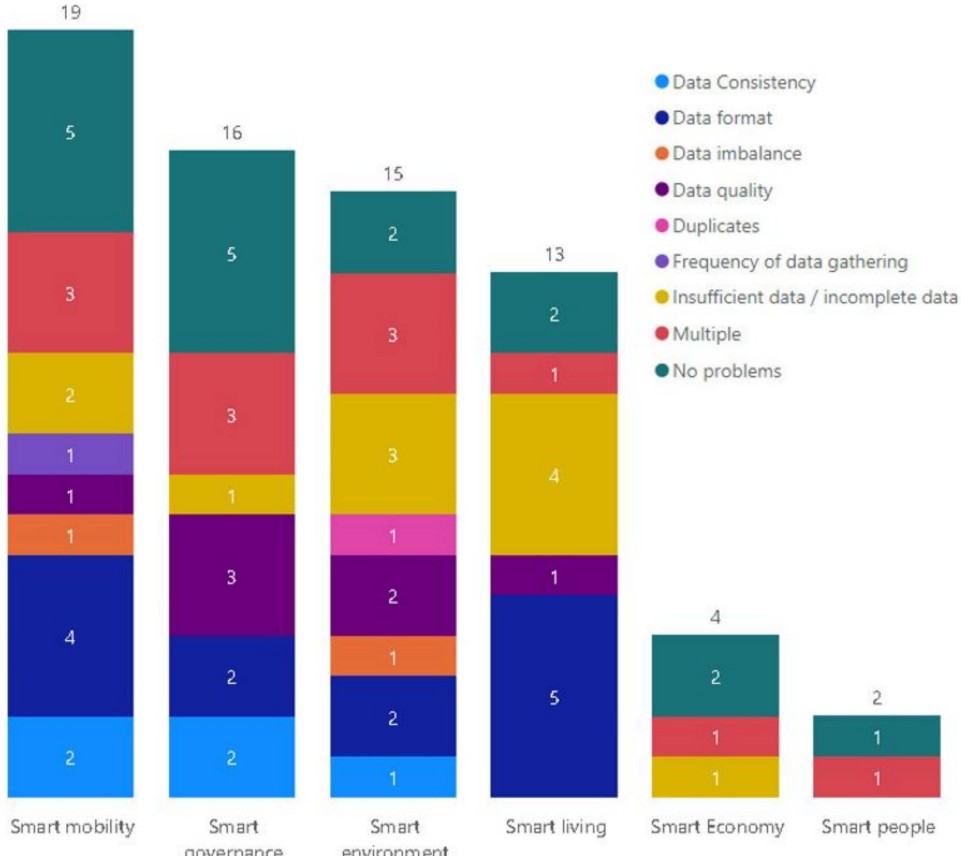

**Figure 18.** Challenges in open data utilization for each of the SC areas of the applications

## 5. Conclusions

Many organizations' leaders today start asking the question 'what can my data do' so beginning to realize the immense potential residing in data. Along with analytics, machine learning already gave the measure of its value and usefulness. However, building and maintaining ML applications is not an easy task and all the experiments, models of application, or case studies in the literature provide an appreciated and beneficial support and inspiration for the smart cities current and forthcoming projects.

AI and machine learning particularly become a core part of businesses, either private or public, around the world. Recent literature [10] pointed out the greater significance of proving the applicability of machine learning techniques in the smart city initiatives. Our research contributes to the current literature on ML applications for SC with a broad analysis that takes into consideration open data, which are becoming more available in the last years. Considering the other SLR approaches, we have decided to delineate and analyze the papers proposing actual applications for SC that make use of ML and open data. This area of investigation proved to be very recent, the initial 472 primary records were published in the last decade (2011–2021), and the final sample of 69 records covers the period of 2013–2021. Furthermore, the increasing number of applications and experiments in the last couple of years denotes the fact that we have addressed an emerging topic.

Some of the most significant results of the executed analysis are listed below:

- The most used source of data is open data platforms (in over 50% of the applications); this result corroborates with [16].
- All the ML techniques were encountered but mostly used are deep learning and classical ML supervised learning. Some ML types have applications in only one SC area (reinforcement learning in smart mobility) and there are dominating choices for others, i.e., deep learning in smart mobility and classical ML supervised learning in

smart governance. All these insights may be useful for upcoming applications because researchers can find direct connections to similar or related applications.

- We proclaim deep learning as the most appealing ML technique because it is exploited in all the SC areas of application, dominating by far in the smart environment and the smart living areas. Deep learning techniques demonstrate that they are able to deal with the huge volumes of data that are constantly produced in modern cities and to develop solutions for the most prevailing urban issues.
- However, the algorithms applied are very diverse, depending on the application and the ML technique. The most employed learning type is the supervised learning and the predominant choice of work is to apply multiple algorithms.
- Among the four types of learning in classical ML applications and in deep learning applications, the supervised learning distinguishes as a preferred option in all the SC areas, based on using open data platforms and operating multiple algorithms.

Furthermore, we have researched the challenges associated with open data utilization in ML applications where our analysis has generated meaningful results. As data can be retrieved from numerous data sources for different domains, there are also issues/challenges for researchers/developers when using these in the SC applications. We discovered that when using an open data platform the challenges could vary from quality of data, to frequency of data collection, to consistency of data, to data format, or to no issues at all. These challenges were described in relation to the different data source categories and the types of ML applications.

Our paper has some limitations. Concerning RQ3, there is a probability of bias in our analysis, while some authors may not have included in their papers the problems they met with open data utilization—sometimes there are not such problems but in other cases they exist but are not stated. We followed PRISMA methodology and then in-depth analysis with visualizations executed with Power BI, but we could also applied, other techniques such as the bibliometric technique to provide more information. Additionally, we could dig more into the ML algorithms usage analysis in order to discover more on the rationale of choosing algorithms for ML applications. From a regional perspective, we did not explore aspects such as the countries or regions that offer the most amount of such papers, the relevance or the irrelevance of election years in contrast with the publishing of such papers, the political climate and open data laws surrounding the regions able to produce the highest amount of such papers. We think about all these shortcomings as opportunities for further research.

This paper has some theoretical and practical implications. The theoretical contribution is that we give a detail perception of the current approaches of open data-based machine learning applications in the smart city initiatives by reviewing the very recent literature. In addition, another theoretical contribution is a valuable synthesis of open data based ML experiments/applications, obtained by classifying the investigated papers by the SC area of application, ML technique and algorithms, and types of open data manipulated. From a practical perspective, in view of the increasing interest from both academia and the industry professionals in researching the SC innovation under the perspective of machine learning and open data huge potential, we believe that they may find valuable insights in our analysis. Our results are particularly helpful for researchers who begin working to develop ML applications for SC because they can calibrate theirs based on previous practical results we have explored in this paper.

**Author Contributions:** Literature review: D.D., A.-M.N., L.H., A.-M.D., F.M.; conceptualization: L.H., D.D., A.-M.D., A.-M.N., F.M.; methodology: D.D. and L.H.; data curation: A.-M.D., A.-M.N., F.M., L.H., D.D.; data processing: F.M., L.H., D.D., A.-M.D., A.-M.N.; validation: L.H. and D.D.; writing: L.H., D.D., A.-M.D., A.-M.N., F.M. All authors have read and agreed to the published version of the manuscript.

**Funding:** This research received no external funding.

**Institutional Review Board Statement:** Not applicable.

**Informed Consent Statement:** Not applicable.

**Conflicts of Interest:** The authors declare no conflict of interest.

## Abbreviations

The following abbreviations are used in this manuscript:

| | |
|---|---|
| AI | Artificial Intelligence |
| ANN | Artificial Neural Network |
| CNN | Convolutional Neural Network |
| DL | Deep Learning |
| ICT | Information and Communication Technology |
| IoT | Internet of Things |
| ITU | International Telecommunication Union |
| LSTM | Long Short-term Memory |
| ML | Machine Learning |
| NN | Neural Network |
| OD | Open Data |
| OGD | Open Government Data |
| SC | Smart City |
| SLR | Systematic Literature Review |

## Appendix A

Table A1. The sample of selected records.

| ID | Authors | Title | Year | Journal/Conference | Publication Details | Scope and Objective | ML Type | SC Area |
|---|---|---|---|---|---|---|---|---|
| [S1] | Abarca-Álvarez, F.J.; Campos-Sánchez, F.S.; Reinoso-Bellido, R. | Demographic and Dwelling Models by Artificial Intelligence: Urban Renewal Opportunities in Spanish Coast | 2018 | International Journal of Sustainable Development and Planning | Vol. 13(7), 941–953 | The purpose of this study is to shed light on the Spanish Mediterranean coast's existing residential models and the relationship with the local demographic reality of users. | Deep learning | Smart governance |
| [S2] | Alamsyah, A.; Gustyana, T.T.; Fajaryanto, A.D.; Septiafani, D. | Open Data Analytical Model for Human Development Index Optimization to Support Government Policy | 2018 | ArXiv | abs/1809.00189 | The case study is on Human Development Index value prediction and its clustered nature, with the purpose of categorizing different level of Human Development Index of cities or region and finding the group characteristics between Human Development Index and GDP. | Unsupervised | Smart governance |
| [S3] | Albert, A.; Kaur, J.; González, M.C. | Using Convolutional Networks and Satellite Imagery to Identify Patterns in Urban Environments at a Large Scale | 2017 | Proceedings of the 23rd ACM SIGKDD International Conference on Knowledge Discovery and Data Mining | 1357–1366 | This paper analyzes patterns in land use in urban neighborhoods using large-scale satellite imagery data and state-of-the-art computer vision techniques based on deep convolutional neural networks. | Supervised | Smart governance |

Table A1. *Cont.*

| ID | Authors | Title | Year | Journal/Conference | Publication Details | Scope and Objective | ML Type | SC Area |
|---|---|---|---|---|---|---|---|---|
| [S4] | Albrecht, C.M.; Zhang, R.; Cui, X.; Freitag, M.; Hamann, H.F.; Klein, L.J.; Finkler, U.; Marianno, F.; Schmude, J.; Bobroff, N.; Zhang, W.; Siebenschuh, C.; Lu, S. | Change Detection from Remote Sensing to Guide Open Street Map Labeling | 2020 | ISPRS International Journal of Geo-Information | Vol.9(7), 427 | The scope of this article is to identifying geospatial regions where mappers need to focus on completing the global OSM dataset. | Supervised | Smart living |
| [S5] | Al-Habashna, A. | Building Height Estimation using Street-View Images, Deep-Learning, Contour Processing, and Geospatial Data | 2021 | 2021 18th Conference on Robots and Vision (CRV) | 103–110 | This paper proposes a ML algorithm and its open-source implementation for automatic estimation of building height from street-view images, using convolutional neural networks (CNNs) and image processing techniques. | Deep learning | Smart living |
| [S6] | Aljohani, N.R.; Fayoumi, A.; Hassan, S.-U. | Predicting At-Risk Students Using Clickstream Data in the Virtual Learning Environment | 2019 | Sustainability | Vol.11(24), 7238 | The proposed study deployed a deep sequential model for the early prediction of at-risk students based on their week-wise clickstream interactions with the VLE. | Deep learning | Smart people |

**Table A1.** *Cont.*

| ID | Authors | Title | Year | Journal/Conference | Publication Details | Scope and Objective | ML Type | SC Area |
|---|---|---|---|---|---|---|---|---|
| [S7] | Awan F.M.; Minerva, R.; Crespi, N. | Improving Road Traffic Forecasting using Air Pollution and Atmospheric data: Experiments based on LSTM Recurrent Neural Networks | 2020 | Sensors | Vol.20(13), 3749 | The article propose a traffic forecasting approach that utilizes air pollution and atmospheric parameters, using data collected from an open data portal. | Deep learning | Smart environment |
| [S8] | Badii, C.; Nesi, P.; Paoli, I. | Predicting Available Parking Slots on Critical and Regular Services by Exploiting a Range of Open Data | 2018 | IEEE Access | Vol.6, 44059–44071 | This paper presents a set of metrics and techniques to predict the number of available parking slots in city garages with gates. | Supervised | Smart mobility |
| [S9] | Barchiesi, D., Preis, T., Bishop, S., and Moat, H.S. | Modelling Human Mobility Patterns using Photographic Data Shared Online | 2015 | Royal Society Open Science | Vol.2, 150046 | The article proposes a model of mobility where displacements are grouped together into geographical clusters. Authors propose a ML algorithm to infer the probability of finding people in geographical locations and the probability of movement between pairs of locations. | Semi-supervised | Smart mobility |

Table A1. *Cont.*

| ID | Authors | Title | Year | Journal/Conference | Publication Details | Scope and Objective | ML Type | SC Area |
|---|---|---|---|---|---|---|---|---|
| [S10] | Barlacchi, G.; Rossi, A.; Lepri, B.; Moschitti, A. | Structural Semantic Models for Automatic Analysis of Urban Areas. | 2017 | Lecture Notes in Computer Science | Vol. 10536, 279–291 | This paper proposes a novel machine learning representation based on the available public information to classify the most predominant land use of an urban area. | Supervised | Smart living |
| [S11] | Barletta, V.S.; Caivano, D.; Nannavecchia, A.; Scalera, M. | A Spell Checking Web Service API for Smart City Communication Platforms | 2019 | Open Journal of Applied Sciences | Vol.9, 819–840 | The goal of the article is to test, through experimental research, the feasibility of the entire project by implementing a spell checking prototype system designed to manage two selected spell checking tools. | Supervised | Smart governance |
| [S12] | Barrera, J.M.; Reina, A.; Maté, A.; Trujillo, J.C. | Solar Energy Prediction Model Based on Artificial Neural Networks and Open Data | 2020 | Sustainability | Vol.12(17), 6915 | The article analyze how the different factors affect the prediction of energy production and how open data can be used to predict the expected output of sustainable sources. | Deep learning | Smart Economy |

**Table A1.** *Cont.*

| ID | Authors | Title | Year | Journal/Conference | Publication Details | Scope and Objective | ML Type | SC Area |
|---|---|---|---|---|---|---|---|---|
| [S13] | Bejarano, G.; Kulkarni, A.; Luo, X.; Seetharam, A.; Ramesh, A. | DeepER: A Deep Learning based Emergency Resolution Time Prediction System | 2020 | International Conferences on Internet of Things (iThings) and IEEE Green Computing and Communications (GreenCom) and IEEE Cyber, Physical and Social Computing (CPSCom) and IEEE Smart Data (SmartData) and IEEE Congress on Cybermatics (Cybermatics) | 490–497 | Authors present a deep learning based emergency resolution time prediction system that predicts future resolution times based on past data. | Deep learning | Smart living |
| [S14] | Buitrago, J.; Asfour, S. | Short-Term Forecasting of Electric Loads Using Nonlinear Autoregressive Artificial Neural Networks with Exogenous Vector Inputs | 2017 | Energies | Vol.10, 40 | The purpose of this study is to develop a more accurate short-term load forecasting method utilizing non-linear autoregressive artificial neural networks with exogenous multi-variable input. | Deep learning | Smart governance |
| [S15] | Chang, Y.; Jang, H. | Traffic Flow Forecast for Traffic with Forecastable Sporadic Events | 2019 | Twelfth International Conference on Ubi-Media Computing (Ubi-Media) | 145–150 | Authors propose an application created to forecast the future traffic flow and plan the driving route. It helps effectively relieve traffic flow, reduce travel time and carbon emissions. | Deep learning | Smart mobility |

Table A1. *Cont.*

| ID | Authors | Title | Year | Journal/Conference | Publication Details | Scope and Objective | ML Type | SC Area |
|---|---|---|---|---|---|---|---|---|
| [S16] | Charan, D.; Teja, D.S.; Subhashini, R.; Jinila, Y.B.; Gandhi, G.M. | Convolutional Neural Network based Water Resource Monitoring Using Satellite Images | 2020 | Fifth International Conference on Communication and Electronics Systems (ICCES) | 1261–1266 | This paper exhibits the extraction of water resources from non-water bodies, for example, vegetation, urban regions etc., using machine learning algorithms. It also targets measuring the area of a particular water body using GIS. | Deep learning | Smart environment |
| [S17] | Chen, C.; Deng, J.; Lv, N. | Illegal Constructions Detection in Remote Sensing Images based on Multi-scale Semantic Segmentation | 2020 | IEEE International Conference on Smart Internet of Things (SmartIoT) | 300–303 | This paper proposes a new convolution structure based on the current semantic segmentation network with the encoding-decoding structure. | Deep learning | Smart governance |
| [S18] | Christantonis, K.; Tjortjis, C.; Manos, A.; Filippidou, D.E.; Mougiakou, E.; Christelis, E. | Using Classification for Traffic Prediction in Smart Cities | 2020 | IFIP Advances in Information and Communication Technology | Vol. 583, 52–61 | This paper relates classification with smart city projects, particularly focusing on traffic prediction. Prediction efforts regard classification tasks aiming at highlighting factors that affect traffic prediction. | Supervised | Smart mobility |

**Table A1.** *Cont.*

| ID | Authors | Title | Year | Journal/Conference | Publication Details | Scope and Objective | ML Type | SC Area |
|---|---|---|---|---|---|---|---|---|
| [S19] | Chung, C.C.; Jeng, T.S. | Information extraction methodology by web scraping for smart cities: Using machine learning to train air quality monitor for smart cities | 2018 | Proceedings of the 23rd International Conference of the Association for Computer-Aided Architectural Design Research in Asia (CAADRIA) | Vol.2, 515–524 | This paper presents an opportunistic sensing system for air quality monitoring to forecast the implicit factors of air pollution. The objective is to develop the Information extraction methodology by web scraping for smart cities. | Supervised | Smart environment |
| [S20] | Cocca, M.; Teixeira, D.; Vassio, L.; Mellia, M.; Almeida, J.M.; Couto da Silva, A.P. | On Car-Sharing Usage Prediction with Open Socio-Demographic Data | 2020 | Electronics | Vol.9, 72–91 | This paper includes a thorough comparison of several machine-learning algorithms in terms of accuracy and ease of training, and also assesses the effectiveness of current state-of-the-art approaches to address the prediction problem. | Supervised | Smart mobility |
| [S21] | Dias, G.M.; Bellalta, B.; Oechsner, S. | Predicting Occupancy Trends in Barcelona's Bicycle Service Stations using Open Data | 2015 | SAI Intelligent Systems Conference | 439–445 | In this work focus, a ML application is created for to make predictions about the statuses of the stations of a public bicycle service. | Supervised | Smart mobility |

**Table A1.** *Cont.*

| ID | Authors | Title | Year | Journal/Conference | Publication Details | Scope and Objective | ML Type | SC Area |
|---|---|---|---|---|---|---|---|---|
| [S22] | Estrada, E., Maciel, R., Ochoa, A., Bernabe-Loranca, B., Oliva, D., Larios, V. | Smart City Visualization Tool for the open data Georeferenced Analysis Utilizing Machine Learning | 2018 | International Journal of Combinatorial Optimization Problems and Informatics | Vol.9 (2), 25–40 | Authors propose a pattern's visualization of efficiency metrics tool, utilizing the auto learning techniques. The objective is to give support to the decision making throughout the georeferenced analysis exploiting the open data. | Unsupervised | Smart people |
| [S23] | Fedorova, S. | GANs for Urban Design. | 2021 | ArXiv | abs/2105.01727 | This paper introduces a new approach towards the context-based generative design of an urban block without a predefined set of constraints, by learning the visual parameters from the existing environment instead and proposes an application of an image-to-image translation GAN to the field of urban design. | Deep learning | Smart living |
| [S24] | Gervasoni, L.; Fenet, S.; Perrier, R.; Sturm, P. | Convolutional Neural Networks for Disaggregated Population Mapping Using Open Data | 2018 | IEEE 5th International Conference on Data Science and Advanced Analytics (DSAA) | 594–603 | In this work, the authors present and evaluate an end-to-end framework for computing disaggregated population mapping employing convolutional neural networks. | Deep learning | Smart governance |

Table A1. *Cont.*

| ID | Authors | Title | Year | Journal/Conference | Publication Details | Scope and Objective | ML Type | SC Area |
|---|---|---|---|---|---|---|---|---|
| [S25] | Gryech, I.; Ben-Aboud, Y.; Guermah, B.; Sbihi, N.; Ghogho, M.; Kobbane, A. | MoreAir: A Low-Cost Urban Air Pollution Monitoring System | 2020 | Sensors | Vol.20, 998 | MoreAir is a low-cost and agile urban air pollution monitoring system. The paper describes the methodology used along with some preliminary data analysis results. | Supervised | Smart environment |
| [S26] | Guo, M.; Wang, P.; Chan, C.; Askary, S. | A Reinforcement Learning Approach for Intelligent Traffic Signal Control at Urban Intersections | 2019 | IEEE Intelligent Transportation Systems Conference (ITSC) | 4242–4247 | In this paper, authors propose a RL approach for traffic signal control at urban intersections. Specifically, they use neural networks as Q-function approximator to deal with the complex traffic signal control problems. | Reinforcement learning | Smart mobility |
| [S27] | Gutierrez, J.M.; Jensen, M.; Henius, M.; Riaz, T. | Smart Waste Collection System Based on Location Intelligence | 2015 | Procedia Computer Science | Vol.61, 120–127 | Authors propose an integrated system model for intelligent waste collection, and the quantification of its benefits and economic costs when deploying and using it for evaluating its feasibility as a real world smart city application. | Unsupervised | Smart environment |

**Table A1.** *Cont.*

| ID | Authors | Title | Year | Journal/Conference | Publication Details | Scope and Objective | ML Type | SC Area |
|---|---|---|---|---|---|---|---|---|
| [S28] | Hebert, A.; Guédon, T.; Glatard, T.; Jaumard, B. | High-Resolution Road Vehicle Collision Prediction for the City of Montreal | 2019 | IEEE International Conference on Big Data (Big Data) | 1804–1813 | This paper shows how one can leverage open datasets of a big city to create high-resolution accident prediction models, authors tested various machine learning methods to deal with the severe class imbalance inherent to accident prediction problems. | Supervised | Smart mobility |
| [S29] | Joglekar, S.; Quercia, D.; Redi, M.; Aiello, L.; Kauer, T.; Sastry, N. | FaceLift: a transparent deep learning framework to beautify urban scenes | 2020 | Royal Society Open Science | Vol.7, 190987 | The purpose of this study is develop a deep learning framework that is able to both beautify existing urban scenes and explain which urban elements make those transformed scenes beautiful. | Deep learning | Smart living |
| [S30] | Kang, S.; Kim, N.; Lee, B. | Fine Dust Forecast Based on Recurrent Neural Networks | 2019 | 21st International Conference on Advanced Communication Technology (ICACT) | 456–459 | This paper proposes a cloud-based NILM system for load monitoring based on the aggregated data such as smart metering into appliance-level load information by using cloud computing and machine learning algorithms implemented in cloud. | Deep learning | Smart environment |

**Table A1.** *Cont.*

| ID | Authors | Title | Year | Journal/Conference | Publication Details | Scope and Objective | ML Type | SC Area |
|---|---|---|---|---|---|---|---|---|
| [S31] | Kee, K.-K.; Lim, Y.S.; Wong, J.; Chua, K.H. | Cloud-based non-intrusive load monitoring system (NILM) | 2019 | Eighth International Conference on Computer Science and Computational Mathematics | | This paper proposed a novel cloud-based NILM application to enable collection of open data for load monitoring and other energy-related services. | Deep learning | Smart Economy |
| [S32] | Kwon, O.; Kim, Y.S.; Lee, N.; Jung, Y. | When Collective Knowledge Meets Crowd Knowledge in a Smart City: A Prediction Method Combining Open Data Keyword Analysis and Case-Based Reasoning | 2018 | Journal of Healthcare Engineering | Vol. 2018 | This paper proposes a case-based reasoning method that combines the use of crowd knowledge from open source data and collective knowledge to diagnose wellness levels in patients suffering from stress or depression. | Semi-supervised | Smart living |
| [S33] | Larsen, G.H.; Yoshioka, L.R.; Marte, C.L. | Bus Travel Times Prediction based on Real-Time Traffic Data Forecast using Artificial Neural Networks | 2020 | International Conference on Electrical, Communication, and Computer Engineering (ICECCE) | 1–6 | The purpose of the paper is to create a methodology to predict the travel times of buses based on open data collected in real-time using an artificial neural network. | Deep learning | Smart mobility |
| [S34] | Li, S.; Zhu, Z.; Wang, H.; Xu, F. | 3D Virtual Urban Scene Reconstruction From a Single Optical Remote Sensing Image | 2019 | IEEE Access | Vol.7, 68305–68315 | This paper presents a low-cost and efficient method for 3D virtual urban scene reconstruction based on multi-source remote sensing big data and deep learning. | Deep learning | Smart living |

Table A1. *Cont.*

| ID | Authors | Title | Year | Journal/Conference | Publication Details | Scope and Objective | ML Type | SC Area |
|---|---|---|---|---|---|---|---|---|
| [S35] | Miao, Z.; Xiao, Y.; Shi, W.; He, Y.; Gamba, P.; Li, Z.; Samat, A.; Wu, L.; Li, J.; Wu, H. | Integration of Satellite Images and Open Data for Impervious Surface Classification | 2019 | IEEE Journal of Selected Topics in Applied Earth Observations and Remote Sensing | Vol.12(4), 1120–1133 | This study sets out to automatically generate training samples from open data, using to classify impervious surface from satellite images. | Supervised | Smart governance |
| [S36] | Miau, S.; Hung, W. H. | River Flooding Forecasting and Anomaly Detection Based on Deep Learning | 2020 | IEEE Access | Vol.8, 198384–198402 | This paper employed a deep learning-based model to predict the water level flood phenomenon. The experimental results showed their proposed method could detect abnormal water levels effectively. | Deep learning | Smart environment |
| [S37] | Milojevic-Dupont; N.; Hans, N.; Kaack, L.; Zumwald, M.; Andrieux, F.; Soares, D.D.; Lohrey, S.; Pichler, P.; Creutzig, F. | Learning from urban form to predict building heights | 2020 | PLoS ONE | Vol.15, 1–22 | This paper presents a machine learning based method for predicting building heights, which is based only on open-access geospatial data on urban form, such as building footprints and street networks. The method allows to predict building heights for regions where no dedicated 3D models exist. | Supervised | Smart governance |

Table A1. *Cont.*

| ID | Authors | Title | Year | Journal/Conference | Publication Details | Scope and Objective | ML Type | SC Area |
|---|---|---|---|---|---|---|---|---|
| [S38] | Mishra, B.K.; Thakker, D.; Mazumdar, S.; Simpson, S.; Neagu, D. | Using Deep Learning for IoT-enabled Camera: A Use Case of Flood Monitoring | 2019 | 10th International Conference on Dependable Systems, Services and Technologies (DESSERT) | 235–240 | This paper explores the use of deep learning in a flood monitoring application in the context of an EC-funded project, smart cities and open data REuse (SCORE). In our work, we apply deep leaning to classify drain blockage images to develop an effective image classification model for different severity of blockages. | Deep learning | Smart environment |
| [S39] | Mishra, B.K.; Thakker, D.; Mazumdar, S.; Neagu, D.; Gheorghe, M.; Simpson, S. | A novel application of deep learning with image cropping: a smart city use case for flood monitoring | 2020 | Journal of Reliable Intelligent Environments | Vol.6, 51–61 | This article proposes a novel image classification approach based on deep learning with an IoT-enabled camera to monitor gullies and drainages. This approach utilizes deep learning to develop an effective image classification model to classify blockage images into different class labels based on the severity. | Deep learning | Smart environment |

**Table A1.** *Cont.*

| ID | Authors | Title | Year | Journal/Conference | Publication Details | Scope and Objective | ML Type | SC Area |
|---|---|---|---|---|---|---|---|---|
| [S40] | Moosavi, V. | Urban morphology meets deep learning: Exploring urban forms in one million cities, town and villages across the planet. | 2017 | ArXiv | abs/1709.02939 | In this work, based on a large data set of street networks in more than one million cities, towns and villages all over the world, authors trained a deep convolutional auto-encoder, that automatically learns the hierarchical structures of urban forms and represents them via dense and comparable vectors. | Unsupervised | Smart governance |
| [S41] | Nassar, A. S.; Lefevre, S. | Automated Mapping of Accessibility Signs With Deep Learning From Ground-level Imagery and Open Data | 2019 | Joint Urban Remote Sensing Event (JURSE) | 1–4 | In this paper, authors aim at detecting accessible parking signs from street view panoramas and geolocalize them, relying on the deep learning object detection methods. | Deep learning | Smart governance |
| [S42] | Nikouei, S. Y.; Chen, Y.; Song, S.; Xu, R.; Choi, B.; Faughnan, T. | Smart Surveillance as an Edge Network Service: From Harr-Cascade, SVM to a Lightweight CNN | 2018 | IEEE 4th International Conference on Collaboration and Internet Computing (CIC) | 256–265 | This work explores the feasibility of two popular human-objects detection schemes, Harr-Cascade and HOG feature extraction and SVM classifier, at the edge and introduces a lightweight Convolutional Neural Network leveraging the depthwise separable convolution for less computation, for human detection. | Deep learning | Smart living |

**Table A1.** *Cont.*

| ID | Authors | Title | Year | Journal/Conference | Publication Details | Scope and Objective | ML Type | SC Area |
|---|---|---|---|---|---|---|---|---|
| [S43] | Paul, U.; Liu, J.; Troia, S.; Falowo, O.; Maier, G. | Traffic-Profile and Machine Learning based Regional Data Center Design and Operation for 5G network | 2019 | Journal of Communications and Networks | Vol.21, 569–583 | This paper is to aid operation and facilitate dynamic utilization of data center resources, using the state-of-the-art recurrent neural network models to predict the future traffic demands according to past demand profiles of each area. | Unsupervised | Smart mobility |
| [S44] | Peppa, M.; Bell, D.; Komar, T.; Xiao, W. | Urban Traffic Flow Analysis based on Deep Learning Car Detection from CCTV Image Series | 2018 | ISPRS—International Archives of the Photogrammetry, Remote Sensing and Spatial Information Sciences | XLII-4, 499–506 | The research aims to analyze traffic flow patterns through fine-tuning pre-trained CNN models on domain-specific low quality imagery, as captured in various weather conditions and seasons. It exploits machine learning algorithms for a wider understanding of traffic congestion and disruption under social events and extreme weather conditions. | Deep learning | Smart mobility |

**Table A1.** *Cont.*

| ID | Authors | Title | Year | Journal/Conference | Publication Details | Scope and Objective | ML Type | SC Area |
|---|---|---|---|---|---|---|---|---|
| [S45] | Pibre, L.; Chaumont, M.; Subsol, G.; Ienco, D.; Derras, M. | How to Deal with Multi-source Data for Tree Detection based on Deep Learning. | 2017 | IEEE Global Conference on Signal and Information Processing (GlobalSIP) | 1150–1154 | This paper focus on managing data from multiple sources for the task of localization of urban trees in multi-source (optical, infrared, DSM) aerial images and then evaluates the different effects of preprocessing on the input data of a CNN. | Deep learning | Smart environment |
| [S46] | Piscini, A.; Romaniello, V.; Bignami, C.; Stramondo, S. A. | A New Damage Assessment Method by Means of Neural Network and Multi-Sensor Satellite Data | 2017 | Applied Sciences | Vol.7(8), 781 | This work presents an application of the ANN inversion technique addressed to the evaluation of building collapse ratio, defined as the number of collapsed buildings with respect to the total number of buildings in a city block, by employing optical and SAR satellite data. | Deep learning | Smart Economy |
| [S47] | Pradhan, I.; Eirinaki, M.; Potika, K.; Potikas, P. | Exploratory Data Analysis and Crime Prediction for Smart Cities | 2019 | Proceedings of the 23rd International Database Applications and Engineering Symposium | 1–9 | The main focus is to apply ML in order to perform an in-depth analysis of the major types of crimes that occurred in the city, observe the trend over the years, and determine how various attributes contribute to specific crimes. | Supervised | Smart governance |

Table A1. *Cont.*

| ID | Authors | Title | Year | Journal/Conference | Publication Details | Scope and Objective | ML Type | SC Area |
|---|---|---|---|---|---|---|---|---|
| [S48] | Castillo-Cara, M.; Rocca, G.B.; Levano, R.A.; Herrera, J.V.; Orozco-Barbosa, L. | Citizen Security using Machine Learning Algorithms through Open Data | 2014 | Proceedings of the 8th IEEE Latin-American Conference on Communications (LATINCOM) | | The paper includes an application proposal based on machine learning algorithms for a possible solution for the public safety problem. The aim of this application is to reduce the threat risk of the physical integrity of pedestrians by geolocating, in real-time, safer places to walk. | Supervised | Smart living |
| [S49] | Roth, J.; Bailey, A.; Choudhary, S.; Jain, R.K. | Spatial and Temporal Modeling of Urban Building Energy Consumption Using Machine Learning and Open Data | 2019 | ASCE International Conference on Computing in Civil Engineering | 459–467 | This paper proposes a new urban building energy model that produces hourly demand profiles for the building stock of New York City using only open publicly available data. | Supervised | Smart Economy |
| [S50] | Scepanovic, S.; Joglekar, S.; Law, S.; Quercia, D. | Jane Jacobs in the Sky: Predicting Urban Vitality with Open Satellite Data | 2021 | ArXiv | abs/2102.00848 | The paper proposes the use of one single source of data, publicly available: Sentinel-2 satellite imagery and tested whether they could automatically extract them with a state-of-the-art deep-learning framework and whether, in the end, the extracted features could predict vitality. | Supervised | Smart living |

**Table A1.** *Cont.*

| ID | Authors | Title | Year | Journal/Conference | Publication Details | Scope and Objective | ML Type | SC Area |
|---|---|---|---|---|---|---|---|---|
| [S51] | Schulz, A.; Ristoski, P.; Paulheim, H. | I See a Car Crash: Real-Time Detection of Small Scale Incidents in Microblogs | 2013 | Lecture Notes in Computer Science | Vol 7955 | This paper presents a solution for a real-time identification of small-scale incidents using micro blogs, thereby allowing to increase the situational awareness by harvesting additional information about incidents. | Supervised | Smart mobility |
| [S52] | Shi, D.; Ding, J.; Errapotu, S. M.; Yue, H.; Xu, W.; Zhou, X.; Pan, M. | Deep Q-Network Based Route Scheduling for Transportation Network Company Vehicles | 2018 | IEEE Global Communications Conference (GLOBECOM) | 44203 | The article evaluates the proposed algorithm's performance via simulations using open data sets from Didi Chuxing. Through extensive simulations, paper show that the proposed scheme is effective in reducing the cruising time of vacant TNC vehicles and improving the earnings of TNC drivers. | Reinforcement learning | Smart mobility |
| [S53] | Shi, D.; Ding, J.; Errapotu, S. M.; Yue, H.; Xu, W.; Zhou, X.; Pan, M. | Deep Q -Network-Based Route Scheduling for TNC Vehicles with Passengers' Location Differential Privacy | 2019 | IEEE Internet of Things Journal | Vol. 6(5), 7681–7692 | This paper propose a deep reinforcement learning based transportation network route scheduling approach, which allows the TNC service center to learn about the dynamic TNC service environment and schedule the routes for the vacant TNC vehicles. | Reinforcement learning | Smart mobility |

**Table A1.** *Cont.*

| ID | Authors | Title | Year | Journal/Conference | Publication Details | Scope and Objective | ML Type | SC Area |
|---|---|---|---|---|---|---|---|---|
| [S54] | Sirmaçek, B. | Remote sensing, AI and innovative prediction methods for adapting cities to the impacts of the climate change | 2021 | ArXiv | abs/2107.02693 | The article propose an AI-based framework which might be useful for extracting indicators from remote sensing images and might help with predictive estimation of future states of these climate adaptation related indicators. | Deep learning | Smart environment |
| [S55] | Spadon, G.; Carvalh A.; Rodrigues, J.F.; Alves, L.G. | Reconstructing commuters network using machine learning and urban indicators | 2019 | Scientific Reports | Vol. 9 | This paper proposes an alternative approach using machine learning and 22 urban indicators to predict the flow of people and reconstruct the intercity commuters' network. | Supervised | Smart mobility |
| [S56] | Stolfi, D.H.; Alba, E.; Yao, X. | Can I Park in the City Center? Predicting Car Park Occupancy Rates in Smart Cities | 2020 | Journal of Urban Technology | 27(7):1–15 | The paper propose a system for collecting public data on car parkoccupancy values, display them in a user-friendly web service, store them to beconsulted as a historical archive, and use these past data to predict the carparks' occupancy rate in the coming week. | Unsupervised | Smart mobility |

**Table A1.** *Cont.*

| ID | Authors | Title | Year | Journal/Conference | Publication Details | Scope and Objective | ML Type | SC Area |
|---|---|---|---|---|---|---|---|---|
| [S57] | Stubbings, P.; Peskett, J.; Rowe, F.; Arribas-Bel, D. | A Hierarchical Urban Forest Index Using Street-Level Imagery and Deep Learning | 2019 | Remote Sens. | Vol.11, 1395 | The article develops a method based on computer vision and a hierarchical multilevel model to derive an Urban Street Tree Vegetation Index which aims to quantify the amount of vegetation visible from the point of view of a pedestrian. | Supervised | Smart environment |
| [S58] | Taran, V.; Gordienko, N.; Kochura, Y.; Gordienko, Y.; Rokovyi, O.; Alienin, O.; Stirenko, S. | Performance Evaluation of Deep Learning Networks for Semantic Segmentation of Traffic Stereo-Pair Images | 2018 | Proceedings of the 19th International Conference on Computer Systems and Technologies | 73–80 | This paper presents the results of application of several deep learning architectures for semantic image segmentation of traffic stereo-pair images from the public cityscapes dataset with a quantitative characterization of the prediction results for the left and right channels (parts) of stereo-pairs. | Deep learning | Smart mobility |
| [S59] | Violos, J.; Pelekis, S.; Berdelis, A.; Tsanakas, S.; Tserpes, K.; Varvarigou, T. | Predicting Visitor Distribution for Large Events in Smart Cities | 2019 | IEEE International Conference on Big Data and Smart Computing (BigComp) | 1–8 | This research examine two sets of supervised machine learning techniques in order to predict the visitors' distribution in the next timesteps and evaluate them using real data from a large music event. | Semi-supervised | Smart governance |

Table A1. *Cont.*

| ID | Authors | Title | Year | Journal/Conference | Publication Details | Scope and Objective | ML Type | SC Area |
|---|---|---|---|---|---|---|---|---|
| [S60] | Vulova, S.; Meier, F.; Fenner, D.; Nouri, H.; Kleinschmit, B. | Summer Nights in Berlin, Germany: Modeling Air Temperature Spatially With Remote Sensing, Crowdsourced Weather Data, and Machine Learning | 2020 | IEEE Journal of Selected Topics in Applied Earth Observations and Remote Sensing | Vol.13, 5074–5087 | In this article, the aim was to predict the spatial distribution of nocturnal air temperature in Berlin, Germany, one day in advance at a 30-m resolution using open-source remote sensing and geodata. | Supervised | Smart environment |
| [S61] | Wang, Y.; Velswamy, K.; Huang, B. | A Long-Short Term Memory Recurrent Neural Network Based Reinforcement Learning Controller for Office Heating Ventilation and Air Conditioning Systems | 2017 | Processes | Vol. 5(3), 46 | In this paper, a model-free actor-critic reinforcement learning controller is designed using long-short-term memory networks. Optimization of thermal comfort alongside energy consumption is the goal in tuning this reinforcement learning controller. | Deep learning | Smart living |
| [S62] | Wieland, M.; Pittore, M. | Performance Evaluation of Machine Learning Algorithms for Urban Pattern Recognition from Multi-spectral Satellite Images. | 2014 | Remote Sens. | Vol. 6, 2912–2939 | The study aims at exploring the potential of machine learning algorithms in the context of an object-based image analysis and to thoroughly test the algorithm's performance undervarying conditions to optimize their usage for urban pattern recognition tasks. | Supervised | Smart governance |

**Table A1.** *Cont.*

| ID | Authors | Title | Year | Journal/Conference | Publication Details | Scope and Objective | ML Type | SC Area |
|---|---|---|---|---|---|---|---|---|
| [S63] | Wu, J.; Zhou, L.; Cai, C.; Dong, F.; Shen, J.; Sun, G. | Towards a General Prediction System for the Primary Delay in Urban Railways. | 2019 | IEEE Intelligent Transportation Systems Conference (ITSC) | 3482–3487 | This paper propose a comprehensive and general data-driven primary delay prediction system framework, which combines general transit feed specification, critical point search, and deep learning models to leverage the data fusion. It also develops an open source data collection and processing tool that reduces the barrier to the use of the different open data sources. | Deep learning | Smart mobility |
| [S64] | Wu, A.N.; Biljecki, F. | Roofpedia: Automatic mapping of green and solar roofs for an open roofscape registry and evaluation of urban sustainability | 2020 | ArXiv | abs/2012.14349 | The scope is to assess the feasibility and accuracy of an automated mapping methodology using multi-step pipeline that combines deep learning and geospatial techniques for detecting sustainable roofs of up to 100% in some cities. | Deep learning | Smart environment |
| [S65] | Xu, X.; Wang, W.; Hong, T.; Chen, J. | Incorporating machine learning with building network analysis to predict multi-building energy use. | 2019 | Energy and Buildings | Vol. 186, 80–97 | The study proposed an interdisciplinary research method to predict multi-building energy use by integrating a social network analysis with an artificial neural network technique. | Deep learning | Smart environment |

**Table A1.** *Cont.*

| ID | Authors | Title | Year | Journal/Conference | Publication Details | Scope and Objective | ML Type | SC Area |
|---|---|---|---|---|---|---|---|---|
| [S66] | Yap, M.; Cats, O. | Predicting disruptions and their passenger delay impacts for public transport stops | 2020 | Transportation | Vol. 48, 1703–1731 | The study objective is to develop a generic approach to predict how often different disruption types occur at different stations of a public transport network, and to predict the impact related to these disruptions as measured in terms of passenger delays. | Deep learning | Smart mobility |
| [S67] | Yokoya, N.; Ghamisi, P.; Xia, J.; Sukhanov, S.; Heremans, R.; Tankoyeu, I.; Bechtel, B. | Open Data for Global Multimodal Land Use Classification: Outcome of the 2017 IEEE GRSS Data Fusion Contest | 2018 | IEEE Journal of Selected Topics in Applied Earth Observations and Remote Sensing | Vol. 5, 1363–1377 | The authors present mixed ideas and methodologies deriving from computer vision and machine learning but also deeply rooted in the specificities of remote sensing for using multidate images and the ensemble methods. | Supervised | Smart governance |
| [S68] | Zhang, P.; Hu, S.; Li, W.; Zhang, C.; Yang, S.; Qu, S. | Modeling fine-scale residential land price distribution: An experimental study using open data and machine learning | 2021 | Applied Geography | Vol. 129, 102442 | This paper attempts to explore the ability of machine learning algorithms to model grid-level of residential land prices using the case of Wuhan in China. Several land price prediction models were built using five machine learning algorithms and various geographic variables. | Supervised | Smart governance |

**Table A1.** *Cont.*

| ID | Authors | Title | Year | Journal/Conference | Publication Details | Scope and Objective | ML Type | SC Area |
|----|---------|-------|------|--------------------|--------------------|--------------------|---------|---------|
| [S69] | Zou, Z.; Ergan, S. | Leveraging Data Driven Approaches to Quantify the Impact of Construction Projects on Urban Quality of Life. | 2019 | ArXiv | abs/1901.09084 | This study provides the details of a machine learning based approach that enables the prediction of impact of construction projects on quality of life in urban settings through the quantification of changes on quality of life indicators (e.g., noise, air quality, traffic) in cities, inferred by open city data. | Unsupervised | Smart living |

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
