# Peer review of "Open Data Based Machine Learning Applications in Smart Cities: A Systematic Literature Review"

_electronics, doi:10.3390/electronics10232997_

Round 1

Reviewer 1 Report

The authors have chosen a relevant area for academic and practical contributions.

I have recommended a major review of this paper, so that the authors can achieve their aimed contributions. The main points are listed below:

1. Abstract

The abstract should provide the reader with the highlights of their work. The main contributions described in the abstract are mainly generic, and should be reviewed.

2. The introduction

The authors should rewrite this section, as there is a lack of cohesion among the paragraphs. This section should tell the reader a story of why are smart cities relevant, and a quick glimpse of the role of machine learning in this context.

There is no need to describe the concept of systematic literature review, as it is quite a common practice in the academic field.

The objective described in the introduction “to investigate all of the academic papers…” is not as limited as it should be, producing further negative impact on the paper.

The ending of the introduction section should provide the reader with a clear indication of the directions where the paper is heading. I suggest the authors rewrite this section with their justification and proper objectives in mind.

I have found it quite interesting to employ PowerBI as a method for the data analyses.

3. Regarding the theoretical background

The first paragraph of this section would be a better fit in the introduction section.

The authors should provide the reader with the concept of smart cities that they have adopted for their paper. This should also be put into a clear statement: eg. “Although SC are often regarded as …, our position is similar to...”

The concepts of Smart Cities, Machine Learning and Open Data could be addressed separately, so the reader would more easily find the topic of interest.

4. In the methods section

There is no need to describe what is a systematic literature review, as it is a common method in academic papers. The authors should focus on further explaining what methods have they adopted to screen the articles, and to perform their data analyses.

I recommend that the research questions should be presented in the introduction section.

The word “strategy” should be replaced with a more usual term, as the authors are describing their research procedures in this section.

5. The conclusions section

Once the objectives are better established, the authors should have more engaging results and more relevant contributions to the chosen field.

Reviewer 2 Report

The work presents a literature review about machine learning methods used to analyse open data related to smart cities applications. The aim is to capture the most relevant trends in the last decade with a focus on type of data sources, type of ML algorithms applied and challenges of this area of application. The method used for the literature review are quite well explained. The results and the comments on them must be improved, both from a graphical point of view and from a descriptive point of view.

  • The introduction must be improved and enriched with some strong motivations for the choice of this topic.
  • At line 53, you considered also papers which analyse Open Data using some machine learning models not related to the world of Smart Cities? If yes, why? The focus of the research is not only applications to Smart Cities?
  • I suggest to shorten section 2, to remove some lists (For example, remove the list of words to describe smart cities in lines 85-88), to add some considerations and make the content more fluid following a specific logic.
  • Figure 2 is not so fundamental for the comprehension, since the explanation in lines 181-189 is sufficient.
  • In lines 278-286, mention the problems of NN (like overfitting) and the advantages of NN (like the flexibility in the application of NN to different type of data)
  • In section Methods explain better you excluded more than 300 papers and which type of papers were removed.
  • Improve the resolution of all the figures and reduce the size of figures. Labels often are difficult to read.
  • In line 355 you stated that the trend is ascending. Why this is your belief, please add some reference and explain better.
  • In Figure 6 add all the years labels in x-axis.
  • In lines 361, you sad that was previously described, where? Point out the reference to the section where you described the areas.
  • In lines 373-376 and also in Figure 8 you separated deep learning methods and supervised models in two different categories. However, often, deep learning models are supervised models, since they predict a target variable, but they can be also unsupervised models. I suggest to explain better these categories, in particular the difference between deep learning, supervised, unsupervised and semi-supervised. If a paper talks about a neural network which predict a target variable, in which category falls? This categorization is misleading.
  • Figure 8 is too large, reduce the size. Add all the labels related to the years in the x-axis and improve the resolution. You can also flip the coordinates.
  • Figure 10 includes two different figures: add a subcaption for each figure or remove the first one since it is the same of Figure 9.
  • Explain better Figure 16, why there are 2 different plots? What is the meaning of the barplot?

Typos

  • Line 41 a developing
  • Line 153 to improve not improving
  • Line 563 Machine without upper M

Round 2

Reviewer 2 Report

The modifications made by the authors are sufficient for the pubblication of the article.